# Learning Harmonic Molecular Representations on Riemannian Manifold

**Yiqun Wang**[1], **Yuning Shen**[1], **Shi Chen**[2]*, **Lihao Wang**[1], **Fei Ye**[1], **Hao Zhou**[3]
[1]ByteDance Research, [2]University of Wisconsin-Madison,
[3]Institute for AI Industry Research (AIR), Tsinghua University
`{yiqun.wang, yuning.shen, wanglihao.1217, yefei.joyce}@bytedance.com,`
`schen636@wisc.edu, zhouhao@air.tsinghua.edu.cn`

## Abstract

Molecular representation learning plays a crucial role in AI-assisted drug discovery research. Encoding 3D molecular structures through Euclidean neural networks has become the prevailing method in the geometric deep learning community. However, the equivariance constraints and message passing in Euclidean space may limit the network expressive power. In this work, we propose a Harmonic Molecular Representation learning (HMR) framework, which represents a molecule using the Laplace-Beltrami eigenfunctions of its molecular surface. HMR offers a multi-resolution representation of molecular geometric and chemical features on 2D Riemannian manifold. We also introduce a harmonic message passing method to realize efficient spectral message passing over the surface manifold for better molecular encoding. Our proposed method shows comparable predictive power to current models in small molecule property prediction, and outperforms the state-of-the-art deep learning models for ligand-binding protein pocket classification and the rigid protein docking challenge, demonstrating its versatility in molecular representation learning.

## 1 Introduction

Molecular representation learning is a fundamental step in AI-assisted drug discovery. Obtaining good molecular representations is crucial for the success of downstream applications including protein function prediction (Gligorijević et al., 2021) and molecular matching, e.g., protein-protein docking (Ganea et al., 2021). In general, an ideal molecular representation should well integrate both *geometric* (e.g., 3D conformation) and *chemical* information (e.g., electrostatic potential). Additionally, such representation should capture features in various *resolutions* to accommodate different tasks, e.g., *high-level* holistic features for molecular property prediction, and *fine-grained* features for describing whether two proteins can bind together at certain interfaces.

Recently, geometric deep learning (GDL) (Bronstein et al., 2017; 2021; Monti et al., 2017) has been widely used in learning molecular representations (Atz et al., 2021; Townshend et al., 2021). GDL captures necessary information by performing neural message passing (NMP) on common structures such as 2D/3D molecular graph (Klicpera et al., 2020; Stokes et al., 2020), 3D voxel (Liu et al., 2021),and point cloud (Unke et al., 2021). Specifically, GDL encodes: a) geometric features by modeling atomic positions in the *Euclidean space*, and b) chemical features by feeding atomic information into the message passing networks. High-level features could then be obtained by aggregating these atom-level features, which has shown promising results empirically.

However, we argue that current molecular representations via NMP in the Euclidean space is not necessarily the optimal solution, which suffers from several drawbacks. First, current GDL approaches need to employ equivariant networks (Thomas et al., 2018) to guarantee that the molecular representations transform accordingly upon rotation and translation (Fuchs et al., 2020), which could undermine the network expressive power (Cohen et al., 2018; Li et al., 2021). Therefore, developing a representation that could properly encode 3D molecular structure while bypassing the equivariance requirement is desirable. Second, current molecular representations in GDL are learned in a

---

*This work was conducted during internship at ByteDance Research.

bottom-up manner, which are hardly able to provide features in different resolutions for different tasks. Specifically, NMP in Euclidean space typically achieves long-range communication between distant atoms by stacking deep layers or increasing the neighborhood radius. This would hinder the effective representation of macromolecules with tens of thousands of atoms (Battiston et al., 2020; Boguna et al., 2021). To remedy this, residue-level graph representations are commonly used for large molecules (Jumper et al., 2021; Gligorijević et al., 2021). Hence designing efficient multi-resolution message passing mechanisms would be ideal for encoding molecules with distinct sizes.

On the other hand, the molecular surface is a high-level representation of a molecule's shape, which has been widely used to study inter-molecular interactions (Richards, 1977; Shulman-Peleg et al., 2004). Intuitively, the interaction between molecules is commonly described as a "key-lock pair", where both shape complementarity (Li et al., 2013) and chemical interactions (e.g., hydrogen bond) determine whether the key matches the lock molecule. It has been shown that the molecular surface holds key information about inter-molecular interactions (Gainza et al., 2020), which makes it an ideal candidate for molecular representation learning (Sverrisson et al., 2021; Somnath et al., 2021).

Inspired by the idea of *Shape-DNA* (Reuter et al., 2006), hereby we propose Harmonic Molecular Representation learning (HMR) by utilizing the Laplace-Beltrami eigenfunctions on the molecular surface (a 2D manifold). Our representation has the following advantages: a) HMR works on 2D Riemannian manifold instead of in the 3D Euclidean space, thus the resulting molecular representation is by design roto-translation invariant; b) HMR represents a molecule in a top-down manner, and is capable of offering multi-resolution features that accommodate various target molecules (i.e., from small molecules to large proteins), thanks to the smooth nature of the Laplace-Beltrami eigenfunctions

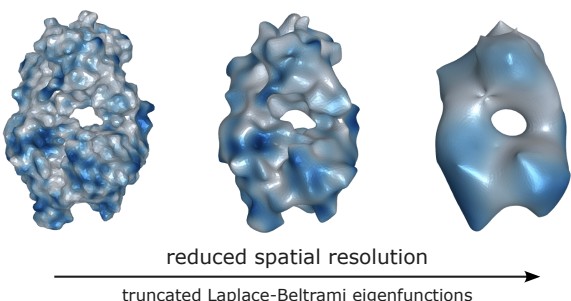

reduced spatial resolution

truncated Laplace-Beltrami eigenfunctions

Figure 1: Multi-resolution molecular surface representation. Showing the electrostatic potential (blue regions being negatively charged, PDB ID: 3V6F) at different resolutions under our representation. See Appendix B for technical details about tuning resolution.

(Fig. 1); c) HMR naturally integrates geometric and chemical features – the molecular shape defines the Riemannian manifold (i.e., the underlying domain equipped with a metric), and the atomic configurations determine the associated functions distributed on the manifold (e.g., electrostatics).

To demonstrate that HMR is generally applicable to different downstream tasks including molecular property prediction and molecular matching, we propose two specific techniques: (1) manifold harmonic message passing for realizing holistic molecular representations, and (2) learning regional functional correspondence for molecular surface matching. Without loss of generality, we apply the proposed techniques to solve three drug discovery-related problems: QM9 small molecule property regression, ligand-binding protein pocket classification, and rigid protein docking pose prediction. Our proposed method shows comparable performance for small molecule property prediction to NMP-based models, while outperforming the state-of-the-art deep learning models in protein pocket classification and the rigid protein docking challenge.

## 2 RELATED WORK

**Molecular Surface Representation**  The molecular surface representation is commonly adopted for tasks involving molecular interfaces (Duhovny et al., 2002), where non-covalent interactions (e.g., hydrophobic interactions) play a decisive role (Sharp, 1994). Non-Euclidean convolutional neural networks (Monti et al., 2017) and point cloud-based learning models (Sverrisson et al., 2022) have been applied to encode the molecular surface for downstream applications, e.g., protein binding site prediction (Mylonas et al., 2021). However, existing methods apply filters with fixed sizes and are highly dependent on the surface mesh quality, which limit the expressive power for molecular shape representation across different spatial scales (Somnath et al., 2021; Isert et al., 2022).

**Geometry Processing**   Our work was inspired by some pioneering work for shape encoding and shape matching in the geometry processing research community (Litman & Bronstein, 2013; Biasotti et al., 2016). The surface of a 3D object is typically discretized into a polygon mesh with vertices and faces. Intrinsic properties of the surface manifold are used to encode the shape (Sun et al., 2009; Bronstein & Kokkinos, 2010). Functional maps (Ovsjanikov et al., 2012; Litany et al., 2017b) have been proposed to establish spectral-space functional correspondence between two manifolds. Recently, deep learning has been applied to learn representative features to facilitate shape recognition and matching (Litany et al., 2017a; Donati et al., 2020; Attaiki et al., 2021).

**Spectral Message Passing**   Our proposed method decomposes surface functions/features as the linear combination of some basis functions (hence the name "harmonic") and realizes message passing by applying various spectral filters. Graph convolutional network (GCN) is closely related to our proposed method, which operates in the graph Laplacian eigenspace (Kipf & Welling, 2016; Shen et al., 2021). The major difference is that graph is discrete by construction, while our method works with continuous Riemannian manifold (Coifman & Lafon, 2006). In other words, the underlying manifold and its spectrum remain the same with different surface discretizations, hence is a robust representation of the surface shape (Coifman et al., 2005). See "Shape-DNA" in Sec. 3.

## 3  PRELIMINARIES

The goal of this work is to propose a representation learning method using the molecular surface, which could properly encode both geometric and chemical features. Intuitively, the molecular surface defines the shape of a molecule in 3D Euclidean space (i.e., geometry), while chemical features (e.g., hydrophobicity) can be treated as functions distributed on the molecular surface. The surface and these associated functions co-determine the underlying molecular properties, e.g., whether an antibody could bind with an antigen. Therefore, we propose to represent a molecule as a set of (learned) functions/features defined on its surface.

Before moving on, we first explain some basic concepts behind our proposed representation learning framework. In Sec. 3.1, we introduce molecular "Shape-DNA" and a set of basis functions inherent to the surface manifold. In Sec. 3.2, we illustrate how to apply harmonic analysis to decompose a function (defined on the molecular surface) into the linear combination of its "Shape-DNA" basis functions, as shown in Fig. 2. These concepts enable us to represent a molecular surface as a 2D Riemannian manifold with associated functions, which form the cornerstone of our proposed framework.

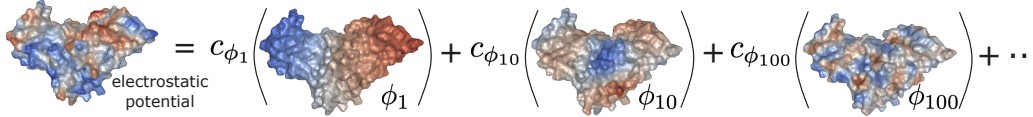

Figure 2: Illustrating manifold harmonic analysis. Left-hand side: the simulated electrostatic potential function on the protein surface. Right-hand side: the linear combination of its Laplace-Beltrami eigenfunctions ($\phi_1, \ldots, \phi_{10}, \ldots$) with corresponding coefficients ($c_{\phi_1}, \ldots, c_{\phi_{10}}, \ldots$). Only a few selected terms are explicitly shown from the infinite sum. Note that these eigenfunctions exhibit different spatial frequencies (resolutions).

### 3.1  THE SHAPE-DNA

The molecular surface [1] can be viewed as a 2D Riemannian manifold ($\mathcal{M}$), which adopts a discrete set of eigenfunctions ($\phi$) that solves

$$\Delta\phi_i = \lambda_i\phi_i, \quad i = 0, 1, \ldots \tag{1}$$

Here, $\Delta$ is the Laplace-Beltrami (LB) operator acting on surface scalar fields, defined as $\Delta f = -\text{div}(\nabla f)$. $\phi_0, \phi_1, \ldots$ are a set of orthonormal eigenfunctions (i.e., $\langle\phi_i, \phi_j\rangle_{\mathcal{M}} = \delta_{ij}$). And $0 = \lambda_0 \leq \lambda_1 \leq \ldots$ are the corresponding eigenvalues. These set of eigenvalues $\{\lambda_i\}$ of the LB operator

---

[1]For instance, an isosurface of its electron density field, or the solvent-accessible/exclusive surface, etc.

are called the "Shape-DNA" and their corresponding eigenfunctions $\{\phi_i\}$ are unique to each shape. [2]. In other words, different molecules adopt different LB eigenfunctions. Notably, two different 3D conformations of the same molecule may not have the same LB eigenfunctions, but a pair of chiral molecules (i.e., mirror image of each other) share the same LB spectrum.

Fig. 2 shows a few selected eigenfunctions ($\phi_1$, $\phi_{10}$, $\phi_{100}$) of a protein surface manifold on the right-hand side. These eigenfunctions are intrinsic properties of the surface manifold, which remain invariant under rigid transformations to the molecule. The eigenvalues $\lambda_i$ reflect the surface Dirichlet energy (defined as $\langle \Delta f, f \rangle_{\mathcal{M}}$), which measures the smoothness of eigenfunction $\phi_i$ over $\mathcal{M}$. The eigenvalues increase linearly, whose slope is roughly inversely proportional to the surface area (known as Weyl's asymptotic law). See Appendix A for more details about the "Shape-DNA".

### 3.2 BASICS OF MANIFOLD HARMONIC ANALYSIS

Now, we introduce basic manifold harmonic analysis, a merit of using the Riemannian manifold representation. "Harmonic analysis" refers to the representation of functions as the superposition of some basic waves. Specifically in our case, given the molecular surface manifold $\mathcal{M}$ and its LB eigenfunctions $\{\phi_i\}_{i=0}^{\infty}$, any scalar-valued function $f$ that is square-integrable on $\mathcal{M}$ can be decomposed into a generalized Fourier series:

$$f(x) = \sum_{i=0}^{\infty} \langle f, \phi_i \rangle_{\mathcal{M}} \phi_i(x). \tag{2}$$

In other words, $f$ can be represented as the linear combination of the LB eigenfunctions. The linear combination coefficient $\langle f, \phi_i \rangle_{\mathcal{M}}$ can be interpreted as the "projection" of $f$ onto the eigenfunction $\phi_i$, which reflects the contribution of this particular eigenfunction to synthesizing function $f$.

Interestingly, it is easy to notice that these eigenfunctions display different spatial frequencies (Fig. 2), i.e., $\phi_1$ varies slowly over the surface, while $\phi_{100}$ oscillates at a much higher frequency. This is analogous to the set of $\sin(kx)$ functions in the 1D case, where high frequency waves with larger $k$ values exhibit more oscillations within the period of length $2\pi$. Hereafter we refer to LB eigenfunctions with (relatively) small/large eigenvalues as the low/high frequency components.

Therefore, by manipulating the linear combination coefficients, we could control the contribution of different frequency components to synthesizing function $f$. For instance, with only low frequency components, the synthesized function will have a lower spatial resolution on the surface (i.e., more smoothed out, Fig. 1 right panel, see Appendix B for technical details.), and vice versa. Synthesizing a new function $f$ with some selected frequency components is commonly known as wave filtering in signal processing, which will be applied in our representation learning framework to realize multi-resolution encoding of the molecular surface. We refer the readers to some excellent review papers for more details about geometry processing (Bronstein et al., 2017; Rosenberg, 1997).

## 4 METHODOLOGY

We realize that the aforementioned geometry processing methods are suitable for molecular surface representation. However, a clear distinction between the 3D objects used in geometry processing research and the molecular systems is that molecules are not simply shapes – their underlying atomic structures beneath the surface govern the molecular functionality. In other words, both geometry (i.e., shape) and chemistry have to be considered for molecular encoding. It has been shown that the molecular surface displays chemical and geometric patterns which fingerprint a protein's mode of interactions with other biomolecules (Gainza et al., 2020). Therefore, we formulate both geometric and chemical properties of a molecule as functions distributed on its surface manifold.

Then comes the question: how do we properly learn these geometric and chemical features on the molecular surface? One viable solution is to emulate the message passing framework commonly used in GNNs, whose goal is to propagate information between distant surface regions to encode surface features at different scales. To that end, we present two methods under the HMR framework. In Sec. 4.1, we introduce manifold harmonic message passing, which makes use of harmonic

---

[2]Two shapes may share the same LB eigendecomposition (isometries). The uniqueness of eigendecomposition up to isometries is associated with the question "Can One Hear the Shape of a Drum?" (Kac, 1966)

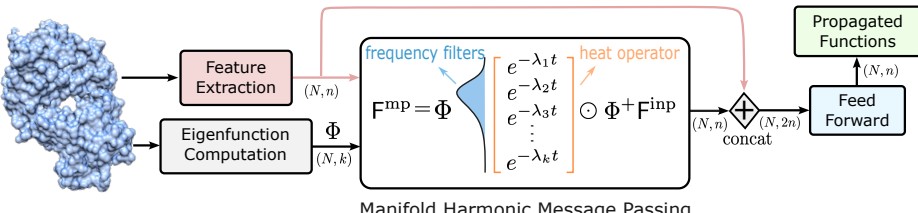

Manifold Harmonic Message Passing

Figure 3: HMR workflow. Given a molecular surface mesh with $N$ vertices, we compute the first $k$ Laplace-Beltrami eigenfunctions (column-wise stacked into an array $\Phi$, and $\Phi^+$ denotes its Moore-Penrose pseudo-inverse, see Appendix B for discrete calculations) with ascending eigenvalues, and extract $n$ initial surface features $\mathsf{F}^{\text{inp}}$ through MLP. Then, we apply neural network-learned spectral filters to propagate the features over the surface to achieve message passing. Note that each feature channel has a unique Gaussian frequency filter and propagation time $t$. Relevant tensor sizes are indicated in parentheses. Multiple message passing blocks can be stacked for better representations.

analysis techniques to allow efficient multi-range communication between regions on the molecular surface regardless of its size. This enables HMR to encode information within a molecule and be applied to molecule-level prediction tasks. In Sec. 4.2, we propose a HMR-powered rigid protein docking pipeline. We use this docking challenge to demonstrate the potential of our proposed representation learning method for modeling interactions between large biomolecules.

## 4.1 LEARNING HARMONIC MOLECULAR REPRESENTATIONS

Given the 3D atomic structure of a molecule, HMR returns (1) a discretized molecular surface manifold (i.e., a triangle mesh) with $N$ vertices $\{\mathbf{x}_1, \ldots, \mathbf{x}_N\} \subset \mathbb{R}^3$ and the corresponding faces; (2) a set of $n$ per-vertex features, $\mathsf{F} \in \mathbb{R}^{N \times n}$, which can be viewed as $n$ learned surface functions. The discretized surface manifold as well as these features represent the geometry and chemistry of the underlying molecule, and can be used for various downstream prediction tasks.

**Surface Preparation** We use MSMS (Ewing & Hermisson, 2010) to compute the molecular solvent-excluded surface as a triangle mesh with $N$ vertices. Then, we compute the first $k$ LB eigenfunctions $\{\phi_i\}_{i \geq 0}^{k-1}$ with ascending eigenvalues as described in Reuter et al. (2009), and stack them into an array $\Phi \in \mathbb{R}^{N \times k}$, where each column stores an eigenfunction.

Geometric features can be readily calculated given the surface mesh. We compute the per-vertex mean curvature, Gaussian curvature, and the Heat Kernel Signatures as described in Sun et al. (2009). Local chemical environment is captured using a simple multilayer perceptron (MLP). For each vertex, we encode its neighboring atoms within a predefined radius (e.g., 6 Å) through MLP, then sum over the neighbors to obtain its chemical embedding. We use another MLP to combine the per-vertex initial features $\mathsf{F}^{\text{inp}} \leftarrow \text{MLP}(\text{concat}(\mathsf{F}^{\text{geom}}, \mathsf{F}^{\text{chem}}))$, $\mathsf{F}^{\text{inp}} \in \mathbb{R}^{N \times n}$. These $n$ features reflect the local geometric and chemical environment of each surface vertex, which will be used as input to the harmonic message passing module. See more implementation details in Appendix C.

The output of the surface preparation module includes (1) the molecular surface triangle mesh, (2) the surface Laplace-Beltrami eigenfunctions $\Phi$, and (3) the per-vertex features $\mathsf{F}^{\text{inp}}$.

**Harmonic Message Passing** Our proposed harmonic message passing mechanism is closely related to the heat diffusion process on an arbitrary surface. Joseph Fourier developed spectral analysis methods to solve the heat equation $\partial f / \partial t + \Delta f = 0$, where $f$ is some heat distributed on the surface. This concise partial differential equation describes how a heat distribution $f$ evolves over time, whose solution can be expressed using the heat operator $\exp(-\Delta t)$, i.e., $f(t) = \exp(-\Delta t)f_0$ for initial heat distribution $f_0$ at $t = 0$. Intuitively, heat will flow from hot regions to cool regions on the surface. As time approaches infinity, the heat distribution $f$ will converge to a constant value (i.e., the global average temperature on the surface), assuming that total energy is conserved.

In fact, heat diffusion can be thought of as a message passing process, where surface regions with different temperatures communicate with each other and propagate the initial heat distribution deterministically. The heat exchange rate is dependent on the difference in temperature (determined by the LB operator), while the message passing distance is determined by the heat diffusion time $t$.

Following this idea, we generalize the heat diffusion process by proposing a function propagation operator $\mathcal{P}$ with neural network-learned frequency filter $F_\theta(\lambda)$:

$$\mathcal{P}f = \sum_i F_\theta(\lambda_i)\langle f, \phi_i \rangle_\mathcal{M} \phi_i \,, \tag{3}$$

$$F_\theta(\lambda) = \exp\left(-\frac{(\lambda - \mu)^2}{\sigma^2}\right) \cdot \exp(-\lambda t) \qquad \text{where} \quad \theta = (\mu, \sigma, t)\,. \tag{4}$$

As shown in Eq. 3, the input function $f$ is first expanded as the linear combination of the LB eigenfunctions with coefficients $\langle f, \phi_i \rangle_\mathcal{M}$. We then learn a spectral-space frequency filter $F_\theta(\lambda)$, which is a function of the corresponding eigenvalue $\lambda$. Here we abuse the usage of frequency, which actually refers to the LB eigenvalues. The filter $F_\theta(\lambda)$ consists of two components (Eq. 4): a Gaussian frequency filter (with parameters $\mu$ and $\sigma$), and the heat operator part $e^{-\lambda t}$ (with parameter $t$). The matrix representation of the function propagation operator $\mathcal{P}$ is plotted in Fig. 3.

For each input function $f$ within $\mathsf{F}^{\mathrm{inp}}$, the neural network learns a unique set of parameters ($\mu$, $\sigma$, $t$) through backpropagation. In Fig. 4, we showcase a few examples of surface functions obtained by applying different frequency filters to an initial delta function.

The Gaussian frequency filter allows the network to propagate the input function along some selected eigenfunctions with eigenvalues close to $\mu$, while the number of selected eigenfunctions is determined by the filter width $\sigma$. Owing to the multi-resolution nature of the LB eigenfunctions with different spatial frequencies, this filter will help capture surface functions at different resolutions.

The heat operator part governs the communication distance. With longer propagation time, function $f$ will be more averaged out towards the global mean, leading to a smoothed function. In addition, the heat operator is by definition a low-pass filter, where components with larger eigenvalues decay faster. This makes eigenfunctions with large eigenvalues contribute less significantly during message passing. Therefore, the combination of the Gaussian frequency filter and heat operator could help the network focus on some higher frequency components (Aubry et al., 2011).

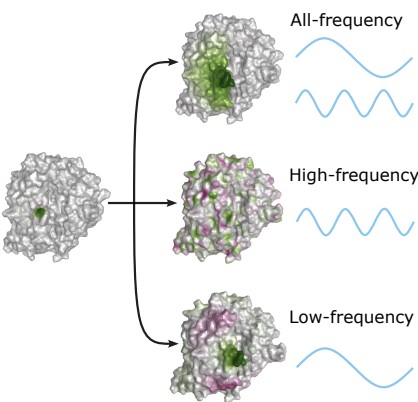

Figure 4: Examples of versatile message passing outcomes using different frequency filter settings.

As demonstrated, HMR is able to represent a variety of surface functions through harmonic message passing. The output of this module has the same size as the input, $\mathsf{F}^{\mathrm{mp}} \in \mathbb{R}^{N \times n}$, where each channel respectively contains the propagated version of its input function. These features represent the neighboring geometric and chemical environment across multiple spatial scales, and can be used for surface property-related tasks, e.g., protein binding site prediction (see Sec. 4.2). In addition, molecule-level representations could be obtained through global pooling (see Sec. 5.1).

## 4.2 Learning Surface Correspondence for Rigid Protein Docking

In this section, we demonstrate how to learn surface correspondence for molecular matching. Specifically, we introduce a surface-based rigid protein docking workflow (Fig. 5) powered by HMR. Rigid protein docking is a significant problem in structural biology, whose goal is to predict the pose of the protein complex based on the structure of the ligand and receptor proteins. It has been shown that protein-protein interfaces exhibit similar geometric and chemical patterns (Gainza et al., 2020). In other words, two proteins may interact if part of their surfaces display similar shapes and chemical functions (i.e., functional correspondence). This is similar to solving a puzzle problem, where both shape and pattern of the missing piece have to match in order to complete the puzzle.

Following this idea, we propose to realize rigid protein docking in two consecutive steps: (1) given two protein surfaces, predict the region where binding might occur (i.e., binding site prediction, locating the missing piece); (2) establish functional correspondence between the ligand/receptor binding surfaces, and convert it to real-space vertex-to-vertex correspondence (shape/pattern matching). Rigid docking could then be achieved by aligning the corresponding binding site surface vertices.

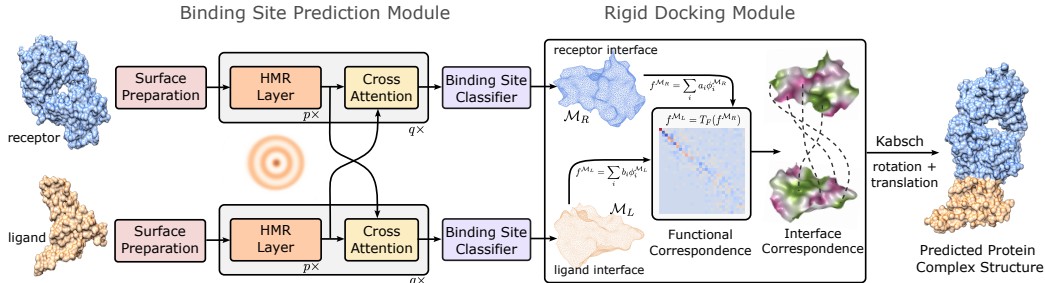

Figure 5: The rigid protein docking pipeline. Given the surface of the ligand and receptor proteins, we apply multiple HMR and cross attention layers to allow communication within and between the surfaces. The learned surface representations are then used to predict the binding interfaces and establish functional correspondence using functional maps. Rigid docking is achieved by converting the functional correspondence to rigid transformations which aligns the predicted interfaces.

**Binding Site Prediction**  Given the ligand and receptor protein surface meshes, we first predict the regions where they interact, which is a per-vertex binary classification problem. We iteratively apply HMR and cross-attention layers (Fig. 5) to encode the surfaces with intra- and inter-surface communications. Next, we use the learned features on each vertex to classify whether it belongs to the binding interface. Detailed descriptions of this module are available in Appendix D.

The output of the binding site prediction module includes: a) the Receptor (Ligand) surface mesh of the binding interface, which is a submanifold of the entire protein surface, represented as $\mathcal{M}_R$ $(\mathcal{M}_L)$ with $N_R$ $(N_L)$ vertices, and b) neural network-learned features or so-called surface functions $\mathsf{F}^{\mathcal{M}_R}$ and $\mathsf{F}^{\mathcal{M}_L}$ distributed on the receptor and ligand binding interfaces, respectively.

**Rigid Docking with Functional Maps**  We know that protein interfaces $\mathcal{M}_R$ and $\mathcal{M}_L$ exhibit similar geometry and also adopt a set of $n$ corresponding functions $\mathsf{F}^{\mathcal{M}_R} \in \mathbb{R}^{N_R \times n}$ and $\mathsf{F}^{\mathcal{M}_L} \in \mathbb{R}^{N_L \times n}$ (Fig. 5). Intuitively, if we could somehow align these set of corresponding functions, then we have found a way to align the protein binding interfaces. To that end, we employ functional maps to "align" $\mathsf{F}^{\mathcal{M}_R}$ and $\mathsf{F}^{\mathcal{M}_L}$ in spectral domain.

Specifically, given the truncated LB eigenfunctions $\Phi^{\mathcal{M}_R} \in \mathbb{R}^{N_R \times k}$ and $\Phi^{\mathcal{M}_L} \in \mathbb{R}^{N_L \times k}$ of the receptor and ligand interface manifolds, we respectively compute the spectral representation of learned functions as $\mathsf{A} = (\Phi^{\mathcal{M}_R})^+ \mathsf{F}^{\mathcal{M}_R}$, and $\mathsf{B} = (\Phi^{\mathcal{M}_L})^+ \mathsf{F}^{\mathcal{M}_L}$, where $+$ denotes the Moore-Penrose pseudo-inverse, and $\mathsf{A}, \mathsf{B} \in \mathbb{R}^{k \times n}$. Functional correspondence ($\mathsf{C}$) can be obtained by minimizing:

$$\mathsf{C} = \underset{\mathsf{C} \in \mathbb{R}^{k \times k}}{\arg\min} \| \mathsf{C}\mathsf{A} - \mathsf{B} \|_{\mathrm{F}},$$

where $\| \cdot \|_{\mathrm{F}}$ denotes the Frobenius norm. Once the functional mapping (i.e., the $\mathsf{C}$ matrix) is recovered through numerical optimization, vertex-to-vertex correspondence between the receptor and ligand surfaces can be established by mapping indicator functions of vertices on $\mathcal{M}_R$ to those on $\mathcal{M}_L$. In practice, we adopted a slightly more complicated functional maps approach, which is illustrated in Appendix E. Finally, we perform rigid docking by aligning the proteins according to the vertex-to-vertex surface correspondence using the Kabsch algorithm (Kabsch, 1976).

## 5  EXPERIMENTS

### 5.1  QM9 MOLECULAR PROPERTY REGRESSION

We employ HMR to perform property regression tasks on the QM9 dataset (Ramakrishnan et al., 2014). We use the same data split and atomic features as Satorras et al. (2021). We compare our results with both invariant and equivariant networks as shown in Table 1, including SchNet (Schütt et al., 2018), NMP (Gilmer et al., 2017), Cormorant (Anderson et al., 2019), SE(3)-Transformer (Fuchs et al., 2020),

Table 1: QM9 Mean Absolute Error

| Task | $\alpha$ | $\Delta\varepsilon$ | $\varepsilon_{\mathrm{HOMO}}$ | $\mu$ | $C_v$ |
|---|---|---|---|---|---|
| Unit | bohr$^3$ | meV | meV | D | cal/(mol K) |
| SchNet | .235 | 63 | 41 | .033 | .033 |
| NMP | .092 | 69 | 43 | .030 | .040 |
| Cormorant | .085 | 61 | 34 | .038 | .026 |
| SE(3)-Tr. | .142 | 53 | 35 | .051 | .054 |
| SEGNN | .060 | 42 | 24 | .023 | .031 |
| **HMR** | .102 | 59 | 37 | .037 | .040 |

and SEGNN (Brandstetter et al., 2021). See Appendix F for the complete table and experimental setup. Interestingly, despite HMR completely discards the bonding information and only performs massage passing over the molecular surface, it still shows comparable performance in predicting these molecular properties.

## 5.2 LIGAND-BINDING POCKET CLASSIFICATION

Next, we showcase the representation power of HMR in predicting the ligand-binding preference of protein pockets. The preference of a protein binding to a specific small molecule (ligand) is associated with the geometric and chemical environment at the binding region (i.e., binding pocket). As defined in Gainza et al. (2020), we classify a protein pocket as one of seven ligand-binding types (ADP, COA, FAD, HEM, NAD, NAP, and SAM) using its surface information. The dataset contains 1,634 training, 202 validation, and 418 test cases. We compare the performance of HMR to MaSIF-ligand (Gainza et al., 2020), which is a geodesic convolutional neural network-based model. A set of features similar to MaSIF-ligand are used in our model: hydrophobicity score and partial charge as chemical features; mean/Gaussian curvature and the Heat Kernel Signatures as geometric features.

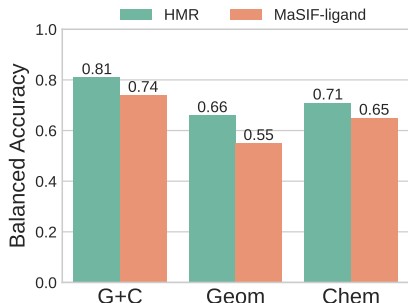

Figure 6: Balanced accuracy of ligand-binding protein pocket classification. Models with different input features are compared: "G+C" includes both geometric and chemical features, while "Geom"/"Chem" only contains geometric/chemical features.

As shown in Fig. 6, the balanced accuracy of our model is consistently better than that reported in Gainza et al. (2020), suggesting that the proposed HMR framework can more effectively encode protein surface information through harmonic message passing over the molecular surface Riemannian manifold. In addition, we draw a similar conclusion that both geometric and chemical information of the binding pockets are important in predicting the type of its binding molecules. See Appendix G for implementation details and more result analysis.

## 5.3 RIGID PROTEIN DOCKING

Finally, we evaluate HMR on a more challenging task: rigid protein docking.

**Experimental setup**  HMR is trained on a modified version of Database of Interacting Protein Structures (DIPS) (Townshend et al., 2019) and evaluated on the gold-standard Docking Benchmark 5.5 (DB5.5) (Guest et al., 2021). We compare our model with the state-of-the-art GNN-based deep learning model EQUIDOCK (Ganea et al., 2021) and two traditional docking methods, ATTRACT (de Vries et al., 2015) and HDOCK (Yan et al., 2020). To evaluate docking performance, we compute the Complex and Interface root-mean-square deviation (RMSD) following Ganea et al. (2021), and calculate DockQ following Basu & Wallner (2016). We also report the success rate indicating whether the result achieves "Acceptable" or higher according to Lensink & Wodak (2013).

**Results**  Model performance are summarized in Table 2 and Fig. 7a. HMR (Top 1) outperforms GNN-based EQUIDOCK model under all metrics. Notably, HMR achieves a much higher success rate, indicating more test cases have results close to the ground truth complex structure. We note that traditional methods still exceed in terms of docking performance but at greater computational

Table 2: Rigid Prediction Results on Docking Benchmark 5.5

| Model | Complex RMSD ↓ | | Interface RMSD ↓ | | DockQ ↑ | | Success rate ↑ | Inference |
| | Median | Mean | Median | Mean | Median | Mean | (≥ Acceptable) | time (sec) |
| --- | --- | --- | --- | --- | --- | --- | --- | --- |
| **HMR (Top 1)** | 12.01 | 12.09 | 10.90 | 11.43 | 0.06 | 0.22 | 0.28 | 5.8 |
| **HMR (Top 3)** | 8.44 | 9.68 | 7.50 | 8.70 | 0.10 | 0.26 | 0.33 | 6.6 |
| EQUIDOCK* | 15.90 | 17.18 | 14.45 | 14.85 | 0.02 | 0.04 | 0.00 | 3.9 |
| ATTRACT | 8.99 | 11.11 | 10.31 | 11.67 | 0.07 | 0.42 | 0.44 | 882.7 |
| HDOCK | 0.39 | 5.97 | 0.32 | 5.68 | 0.97 | 0.71 | 0.73 | 884.9 |

*EQUIDOCK model is trained on DIPS, provided by the authors

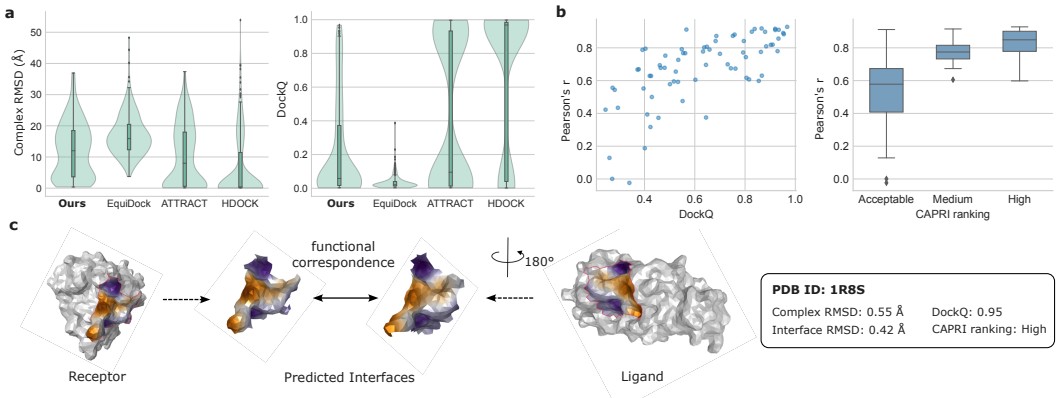

Figure 7: Rigid docking performance analysis. **a**. Distribution of Complex RMSD and DockQ scores for poses predicted using HMR (ours, Top 1), EQUIDOCK, HDOCK, and ATTRACT. **b**. Correlations between learned functions on the predicted ligand-receptor interfaces. For each test case, we calculate the correlation as the Pearson's $r$ between the learned function values on ligand interface vertices and their nearest receptor vertices (corresponding vertices), averaged over 128 hidden channels. **c**. A showcase of functional correspondence between interfaces for a well-docked case.

cost (Table H.1). Further experiments show that the harmonic message passing mechanism learns to propagate information at different scales (Fig. H.1) and is critical to the effectiveness of the model (Table H.2). Our feature ablation studies show that chemical information is particularly important in the rigid protein docking task (Table H.3). Compared to EQUIDOCK, the novel framework of HMR and optimized training dataset collaboratively contribute to the higher performance (Table H.4).

We further examine the learned features at the predicted binding interfaces. As shown in Fig. 7bc, surface functions on the two binding sites are highly correlated, confirming the good alignment achieved using functional maps. Cases with higher docking quality show stronger interface feature correlations, suggesting our HMR is able to learn complex surface interaction patterns, which also supports the claims in Gainza et al. (2020) about the significant role of protein surface.

HMR predicts multiple binding sites for some proteins (either due to certain protein symmetries or model uncertainty). Therefore, we also assess the model performance by including candidate poses from top 3 binding site pairs, ranked by the mean probability predicted by the binding site classifer. The best scores from top 3 poses show that the performance of HMR is competitive to ATTRACT. See Appendix H for more detailed analysis on the rigid protein docking experiment.

## 6 CONCLUSIONS AND OUTLOOK

We presented HMR, a powerful surface manifold-based molecular representation learning framework. By integrating geometric and chemical properties as functions distributed on the molecular surface manifold, and applying harmonic message passing in spectral domain, we achieve multi-resolution molecular representations. HMR shows promising performance in molecular property prediction, protein pocket classification, and molecular matching tasks. Our work highlights an important aspect of molecular "structure-activity" relationship, that is – "shape-activity" relationship. We think this is particularly significant for large biomolecules, where the surface shape and chemical patterns determine some fundamental biological activities, such as protein-protein interactions.

The HMR framework could serve as a complementary method to GNN/NMP-based models in solving challenges for complex biological systems, which exhibits its unique advantages and shortcomings. One of the foreseeable challenges in further developing the HMR framework is the efficient computation of the "Shape-DNA" (i.e., solving a second-order partial differential equation) in order to incorporate protein dynamics. Since either a change of the protein backbone or some side chains near the surface may alter the entire "Shape-DNA", which needs to be recomputed upon protein conformational change. This is particularly important for large biomolecules where the surface shape undergoes significant changes (e.g., the Complementarity-Determining Regions of an antibody upon binding with an antigen, or the allosteric site of some enzymes). To that end, we call for more research attention to surface manifold-based molecular representation learning.

## ACKNOWLEDGEMENTS

The authors thank Dr. Hang Li and Dr. Quanquan Gu for their insightful comments. Hao Zhou is supported by Vanke Special Fund for Public Health and Health Discipline Development, Tsinghua University (NO.20221080053), Guoqiang Research Institute General Project, Tsinghua University (No. 2021GQG1012).

## REPRODUCIBILITY STATEMENT

The code and data are available at https://github.com/GeomMolDesign/HMR. QM9 raw dataset is provided at `https://springernature.figshare.com/ndownloader/files/3195389`. The dataset for the ligand-binding pocket classification is provided at `https://zenodo.org/record/2625420` and the split used by MaSIF is at `https://github.com/LPDI-EPFL/masif/tree/master/data/masif_ligand/lists`. DIPS dataset can be downloaded from the following website `https://github.com/BioinfoMachineLearning/DIPS-Plus`. EQUIDOCK model and checkpoints can be downloaded from `https://github.com/octavian-ganea/equidock_public`. ATTRACT can be downloaded from `https://github.com/sjdv1982/attract`. HDOCK is implemented using its local version `HDOCKlite`, which can be downloaded from `http://huanglab.phys.hust.edu.cn/software/hdocklite/`. DockQ can be downloaded from `https://github.com/bjornwallner/DockQ/`.

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

# Appendix

## A  THE MOLECULAR SHAPE-DNA

**Riemannian Manifold**  A manifold is a space that is locally flat but not necessarily globally flat. Formally speaking, a $d$-dimensional manifold $\mathcal{M}$ is a topological space where each point $p \in \mathcal{M}$ has a neighborhood that is homeomorphic to a $d$-dimensional Euclidean space (Lee, 2013), which is equivalent to the tangent space at $p$ and is denoted by $T_p\mathcal{M}$.

We can further assign a positive definite inner product $g : T_p\mathcal{M} \times T_p\mathcal{M} \rightarrow \mathbb{R}$ on every tangent space, and this inner product is called a Riemannian metric. A manifold equipped with a Riemannian metric is called a Riemannian manifold. Intuitively, the Riemannian metric provides a measurement of the velocity when a particle moves on the manifold, and many other quantities can therefore be defined. For example, for any tangent vector $X_p \in T_p\mathcal{M}$, the quantity $|X_p| := \sqrt{g(X_p, X_p)}$ can be interpreted as the traveling speed of a particle when passing through $p$. Hence, the traveling distance along a curve (i.e., the length of a curve) can be defined as the integral of $|X_p|$ along the curve. On a Riemannian manifold, quantities that can be expressed in terms of the Riemannian metric are called intrinsic (e.g., geodesic distance).

When a manifold is realized in the Euclidean space, a natural Riemannian metric can be induced from the ambient Euclidean space. We always refer to this induced metric when talking about a Riemannian manifold in the following appendices.

**The Lalpace-Beltrami Operator**  We denote a real-valued scalar function on the manifold $\mathcal{M}$ by $f$. Given two functions $f_1, f_2$ on the manifold, we can define the inner product $\langle f_1, f_2 \rangle_{\mathcal{M}} = \int_{\mathcal{M}} f_1(x) f_2(x) \mathrm{d}\mu(x)$, where the area element $\mathrm{d}\mu$ is induced by the Riemannian metric. We denote by $L^2(\mathcal{M}) = \{f : \mathcal{M} \rightarrow \mathbb{R} \mid \langle f, f \rangle_{\mathcal{M}} < \infty\}$ the space of square-integrable functions on $\mathcal{M}$. On a Riemannian manifold $\mathcal{M}$, we can generalize the usual Euclidean gradient $\nabla f$ and the positive semi-definite Laplace operator $\Delta f = -\mathrm{div}(\nabla f)$ to the intrinsic gradient $\nabla_{\mathcal{M}} f$ and the Laplace-Beltrami (LB) operator $\Delta_{\mathcal{M}} f$, respectively (Petersen, 2006).

The LB operator admits a discrete set of eigenfunctions that solves

$$\Delta_{\mathcal{M}} \phi_i(x) = \lambda_i \phi_i(x) \qquad x \in \mathcal{M} \tag{A.1}$$

with homogeneous Neumann boundary conditions if $\mathcal{M}$ has a boundary. Here, $0 = \lambda_1 \leq \lambda_2 \leq \ldots$ are eigenvalues and $\phi_1, \phi_2, \ldots$ are the corresponding eigenfunctions. The LB eigenfunctions form an orthonormal basis of $L^2(\mathcal{M})$, i.e., $\langle \phi_i, \phi_j \rangle_{\mathcal{M}} = \delta_{ij}$.

**Molecular Shape DNA**  Intuitively, the LB eigenfunctions are the smoothest functions on the surface, where smoothness is measured by the Dirichlet energy ($\langle \Delta f, f \rangle_{\mathcal{M}}$). Eigenfunctions with smaller eigenvalues are more smooth (i.e., with lower spatial frequency). However, these eigenfunctions are not directly comparable across different manifolds. Fig. A.1a shows the first 300 eigen-

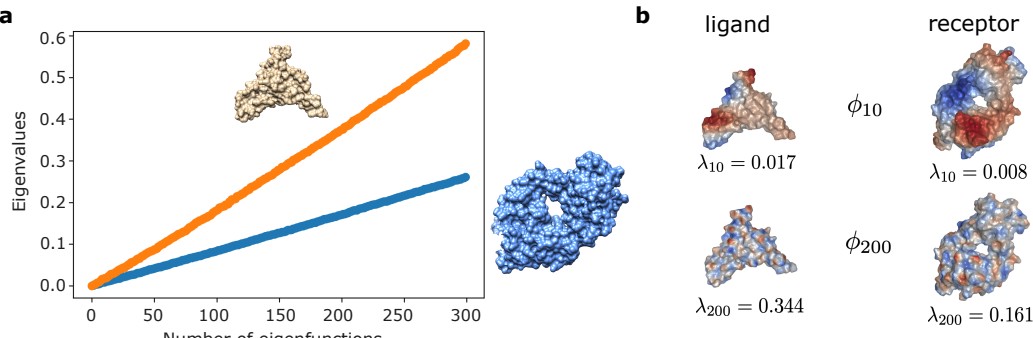

Figure A.1: **a** Weyl's asymptotic law, where the protein with larger surface area exhibits a slower eigenvalue growth rate. **b** The $10^{\text{th}}$ and $200^{\text{th}}$ eigenfunction of the ligand and receptor surface manifold, respectively. Eigenfunctions with similar eigenvalues exhibit similar spatial frequencies regardless of the surface area, since they have similar smoothness in the sense of Dirichlet energy.

values of a pair of ligand and receptor protein surfaces (PDB ID: 3V6Z). The protein with smaller surface area exhibits a larger slope, known as Weyl's asymptotic law. Fig. A.1b presents a few eigenfunctions with correspondending eigenvalues for the ligand and receptor molecules. Since different eigenfunctions have their unique spatial resolutions, in our experiments we compute all eigenfunctions with eigenvalues smaller than a predefined value (determined empirically) to guarantee that molecules of different sizes have the same highest spatial resolution in their eigenfunctions.

# B  IMPLEMENTING MANIFOLD HARMONIC ANALYSIS

**Laplace-Beltrami Eigendecomposition**  Given a Riemannian manifold $\mathcal{M}$ and its Laplace-Beltrami operator $\Delta_{\mathcal{M}}$, the Laplacian eigenvalue problem we consider states as

$$\Delta_{\mathcal{M}} f = \lambda f,$$

with homogeneous Neumann boundary condition.

To realize discrete calculations, we approximate the manifold with a triangle mesh consisting of $N$ vertices $\{\mathbf{x}_i\}_{i=1}^{N} \subset \mathcal{M}$, and the corresponding faces. We then solve the discretized eigenvalue problem using a linear finite element method (FEM) (Reuter et al., 2009). The discretized equation we obtain is the following generalized eigenvalue problem

$$\mathsf{A}_{\mathrm{cot}}\mathsf{f} = \lambda \mathsf{B}\mathsf{f}, \quad \mathsf{f} := (f(\mathbf{x}_i))_{i=1}^{N} , \tag{B.1}$$

where $\mathsf{A}_{\mathrm{cot}}$ is the stiffness matrix with cotagent weights,

$$\mathsf{A}_{\mathrm{cot}}(i,j) := \begin{cases} \frac{1}{2}(\cot \alpha_{ij} + \cot \beta_{ij}) & \text{if } (i,j) \text{ is an edge} \\ -\sum_{k \in \mathcal{N}(i)} A_{\mathrm{cot}}(i,k) & \text{if } i = j \\ 0 & \text{otherwise}, \end{cases}$$

and $\mathsf{B}$ is an $N \times N$ sparse mass matrix which is associated with the weight of each surface vertex,

$$\mathsf{B}(i,j) := \begin{cases} \frac{1}{12}(|t_1| + |t_2|) & \text{if } (i,j) \text{ is an edge} \\ \frac{1}{6}\sum_{t_k \in T(i)} |t_k| & \text{if } i = j \\ 0 & \text{otherwise}. \end{cases}$$

Here $\alpha_{ij}$ and $\beta_{ij}$ are the two angles opposite to the edge $(i,j)$, and $\mathcal{N}(i)$ denotes the vertices that are adjacent to vertex $i$. The set $T(i)$ contains all the triangles that have $i$ as its vertex, and $|t_i|$ is the area of the triangle $t_i$. We also use $t_1$ and $t_2$ to denote the triangles that share the edge $(i,j)$.

Such generalized symmetric eigenvalue problem can be solved with commonly used numerical simulaiton packages (e.g., `scipy`). We can find non-negative eigenvalues $\Lambda$ and eigenvectors $\mathsf{Z}$ such that

$$\mathsf{Z}^{\top}\mathsf{A}_{\mathrm{cot}}\mathsf{Z} = \Lambda, \quad \mathsf{Z}^{\top}\mathsf{B}\mathsf{Z} = \mathsf{I}.$$

$\mathsf{I}$ is the identity matrix, $\Lambda := \mathrm{diag}(\lambda_0, \lambda_1, \dots)$ is the diagonal matrix of the eigenvalues, and the matrix $\mathsf{Z} \in \mathbb{R}^{N \times N}$ has the eigenvectors of Eq. B.1 as its column vectors. Note that, unlike other conventional orthonormal basis (i.e., $\mathsf{Q}^{\top}\mathsf{Q} = \mathsf{I}$), here $\mathsf{Z}$ forms an orthonormal basis w.r.t. the mass matrix $\mathsf{B}$, that is, $\mathsf{Z}^{\top}\mathsf{B}\mathsf{Z} = \mathsf{I}$.

Practically, we do not need to store the entire $N \times N$ eigenvector matrix $\mathsf{Z}$. Just like in Fourier series expansion, a truncated Fourier basis with finite terms can be used to approximate the original signal. The number of basis that we keep determines the resolution of the synthesized signal, where typically high-frequency components are truncated. In our case, we empirically determine the number of eigenvectors to keep for different molecular systems, which is also task-dependent.

**Resolution Tuning with Harmonic Analysis**   We now explain how to realize surface resolution tuning under our representation framework (e.g., how to make Fig. 1).

Given a molecular surface triangle mesh, we first compute its Laplace-Beltrami eigendecomposition as described above. Under the discrete setting, we refer to the eigenfunctions as eigenvectors. We obtain a set of truncated Laplace-Beltrami eigenvectors $\mathsf{Z} \in \mathbb{R}^{N \times k}$ ($k$ is the number of eigenvectors we keep), the associated $k$ eigenvalues $\{\lambda_i\}_{i=0}^{k-1}$, and the sparse mass matrix $\mathsf{B} \in \mathbb{R}^{N \times N}$.

Let $\mathsf{f} \in \mathbb{R}^N$ be the initial function of interest (stored as an $N$-dimensional array), which is distributed on the underlying molecular surface (e.g., electrostatic potential, a scalar value at each surface vertex). The spectral representation of function $\mathsf{f}$ can be calculated as

$$\mathsf{f}^{\mathrm{spec}} = \mathsf{Z}^\top \mathsf{B} \mathsf{f}, \quad \mathsf{f}^{\mathrm{spec}} \in \mathbb{R}^k,$$

which is similar to a discrete Fourier transform. To project the function back to real space (inverse Fourier transform), we simply do

$$\mathsf{f}' = \mathsf{Z} \mathsf{f}^{\mathrm{spec}}, \quad \mathsf{f}' \in \mathbb{R}^N.$$

Resolution tuning is achieved by controlling the number of basis we use (i.e., tuning $k$) in spectral space. As shown in Fig. 1, the electrostatic potential function resolution can be tuned by manipulating the number of Laplace-Beltrami eigenfunctions.

However, what about the resolution of the shape itself? In Fig. 1, we see that the surface smoothness can also be tuned. It is important to realize that manifold is an *abstract* concept, which does not necessarily have a particular *realization* in the Enclidean space. The surfaces that we visualize in Fig. 1 are realizations of the underlying manifold in the Euclidean space, where each surface vertex is associated with some Cartesian coordinates. These coordinates are extrinsic properties of the manifold, thus can be treated in the same way as other surface functions (e.g., the electrostatic potential). Therefore, in order to reconstruct the molecular surface with a lower spatial resolution, we can simply calculate the smoothed Cartesian coordinates:

$$\mathsf{x}' = \mathsf{Z} \mathsf{x}^{\mathrm{spec}} = \mathsf{Z} \mathsf{Z}^\top \mathsf{B} \mathsf{x},$$

and do the same for $\mathsf{y}$ and $\mathsf{z}$ to obtain the smoothed surface coordinates $(\mathsf{x}', \mathsf{y}', \mathsf{z}')$. In short, we realize resolution tuning with a series of (sparse) matrix multiplications bringing the surface functions back-and-forth between the real space and the generalized Fourier space.

## C   SURFACE PREPARATION

The raw input to the HMR framework is simply the 3D atomic structures (e.g., `xyz` files for small molecules, or `PDB` files for proteins). In other words, we only need to know where these atoms are and their atomic species. For proteins with only heavy atoms (since hydrogen atoms are almost invisible under X-ray diffraction detectors), we use `reduce` (Word et al., 1999) (or alternatively `PDB2PQR` (Dolinsky et al., 2004)) software to add hydrogen atoms.

Next, we employ `MSMS` (Ewing & Hermisson, 2010) to calculate the solvent-excluded surface of the molecule (with probe radius 1.5 Å, sampling density 3.0 for small molecules and 1.0 for proteins) as a triangle mesh. We use `PyMesh` (Zhou, 2019) to further refine the surface mesh in order to reduce the number of vertices and fix poorly meshed areas. Degenerate vertices or disconnected surfaces would lead to numerical issues for solving the generalized eigenvalue problem in the next step, thus should be fixed beforehand. We then compute the truncated Laplace-Beltrami eigenvectors, eigenvalues, and the mass matrix as described in Appendix B.

Initial geometric features can be directly calculated given the surface triangle mesh, where we use the `libigl` (Jacobson & Panozzo, 2017) package to calculate the mean and Gaussian curvatures, and compute the Heat Kernel Signatures as described in Sun et al. (2009). These geometric features capture shape-related properties of the molecular surface, and are stored as a scalar-type array $\mathsf{F}^{\mathrm{geom}} \in \mathbb{R}^{N \times p}$, where $p$ is the number of initial geometric features (a user defined variable).

Chemical features are projected from atoms to their neighboring surface vertices. We first obtain an initial descriptor vector $\mathbf{u}$ for each atom (e.g., atomic number, charge, etc.). Then, for each surface

vertex $\mathbf{x}_i$, we compute its $\nu$ nearest neighbor atoms centered at $\{\mathbf{a}_1^i, \ldots, \mathbf{a}_\nu^i\}$ with features $\{\mathbf{u}_1^i, \ldots, \mathbf{u}_\nu^i\}$. We apply a multilayer perceptron (MLP) to the vector $[\,\mathbf{u}_\nu^i, 1/\|\mathbf{x}_i - \mathbf{a}_\nu^i\|\,]$ for each neighboring atom, then compute the average over the neighbors to obtain the chemical feature vector $\mathsf{F}^{\text{chem}} \in \mathbb{R}^{N \times q}$, where $q$ is a user defined variable indicating the dimension of initial chemical features. In short, the initial chemical features of each surface vertex are determined by its neighboring atomic species and their distance, which is learned by a MLP.

We use another MLP to combine the per-vertex initial features

$$\mathsf{F}^{\text{inp}} \leftarrow \text{MLP}(\text{concat}(\mathsf{F}^{\text{geom}}, \mathsf{F}^{\text{chem}})), \quad \mathsf{F}^{\text{inp}} \in \mathbb{R}^{N \times n}.$$

These $n$ features reflect the local geometric and chemical environment of each surface vertex, which will be used as the input to the harmonic message passing module.

The output of the surface preparation module includes (1) the truncated Laplace-Beltrami eigenvectors $\mathsf{Z} \in \mathbb{R}^{N \times k}$, the corresponding eigenvalues $\{\lambda_i\}_{i=0}^{k-1}$, and the sparse mass matrix $\mathsf{B} \in \mathbb{R}^{N \times N}$, (2) per-vertex scalar features $\mathsf{F}^{\text{inp}} \in \mathbb{R}^{N \times n}$.

## D  THE BINDING SITE PREDICTION MODULE

First, we define the binding site as the protein surface region which is within 3 Å to its counterpart surface, and obtain the set of all corresponding surface points $P$ as the nearest neighbor vertices between the ground truth protein interfaces.

The binding site prediction module stacks two feature propagation blocks, each consists of three HMR layers (introduced in Sec. 4.1) and a cross attention layer. Given the propagated receptor features $\mathsf{F}$ and ligand features $\mathsf{G}$, the cross attention layer enables communication between proteins:

$$\mathsf{F}' = \text{softmax}\left(\frac{(\mathsf{F}\mathsf{W}_\text{Q})(\mathsf{G}\mathsf{W}_\text{K})^\top}{\sqrt{d_k}}\right)(\mathsf{G}\mathsf{W}_\text{V}),$$

$$\mathsf{G}' = \text{softmax}\left(\frac{(\mathsf{G}\mathsf{W}_\text{Q})(\mathsf{F}\mathsf{W}_\text{K})^\top}{\sqrt{d_k}}\right)(\mathsf{F}\mathsf{W}_\text{V}),$$

where $\mathsf{F} \in \mathbb{R}^{N_R \times d_k}$, $\mathsf{G} \in \mathbb{R}^{N_L \times d_k}$ denotes the propagated features on the receptor/ligand protein surface, $d_k$ denotes the dimension of features, and $\mathsf{W}_\text{Q}$, $\mathsf{W}_\text{K}$ and $\mathsf{W}_\text{V}$ are the parameter matrices for the query, key, and value in attention computation, respectively.

The loss function consists of two components. The first is a binary cross entropy loss, which encourages the model to correctly predict the binding site:

$$\mathcal{L}_{bce}(i) = -[y_i \log x_i + (1 - y_i) \log(1 - x_i)],$$

where $y_i$ and $x_i$ are the label and predicted probability of whether vertex $i$ belongs to the binding site.

The second term is a PointInfoNCE loss (Xie et al., 2020), a contrastive matching loss that minimizes the distance between the features of corresponding surface points and maximizes the distance between non-corresponding point features:

$$\mathcal{L}_{match} = -\sum_{(i,j) \in P} \log \frac{\exp(\mathbf{f}_i \cdot \mathbf{g}_j / \tau)}{\sum_{(\cdot, k) \in P} \exp(\mathbf{f}_i \cdot \mathbf{g}_k / \tau)},$$

where $P$ is the set of all corresponding surface points and $\tau$ is the temperature factor (a hyperparameter). Here $\mathbf{f}_i$ and $\mathbf{g}_j$ are the neural network-learned feature vectors at point $i$ and $j$, which belong the the receptor and ligand surface, respectively.

The total loss is the weighted sum of the two loss terms:

$$\mathcal{L} = \mathcal{L}_{bce} + \lambda \mathcal{L}_{match},$$

where empirically we set $\lambda$ to 0.1 in our docking experiments.

# E  DETAILS ON FUNCTIONAL MAPS

In this section, we present the details of functional maps used in Sec. 4.2.

Let us be given two manifolds $\mathcal{M}$ and $\mathcal{N}$. The aim of functional maps is to find a bijective mapping $T : \mathcal{M} \to \mathcal{N}$ to align these two manifolds. Unlike traditional methods that try to recover the point-to-point correspondence directly, functional maps lift the mapping $T$ to a correspondence between the functional spaces on the two manifolds. Formally, let $L^2(\cdot)$ be the functional space of square integrable functions on a manifold, we infer the functional correspondence $T_F : L^2(\mathcal{M}) \to L^2(\mathcal{N})$ induced by the mapping $T$, and is defined by $T_F f = f \circ T^{-1}$ for any $f \in L^2(\mathcal{M})$.

To compute the functional correspondence, we need to utilize the Laplace-Beltrami basis on the manifolds. Actually, such functional correspondence has a concise expression in spectral domain: given the respective truncated LB eigenfunctions $\{\phi_j^{\mathcal{M}}\}_{j\geq 0}^{k_1}$ on $\mathcal{M}$ and $\{\phi_i^{\mathcal{N}}\}_{i\geq 0}^{k_2}$ on $\mathcal{N}$, the functional correspondence $T_F$ can be (approximately) represented as a change of basis matrix:

$$\mathsf{C} = (c_{ij})_{k_2 \times k_1} = (\langle \phi_i^{\mathcal{N}}, T_F \phi_j^{\mathcal{M}} \rangle_{\mathcal{N}})_{k_2 \times k_1}.$$

Now given a set of $q$ corresponding functions $\{f_1, \ldots, f_q\} \subset L^2(\mathcal{M})$ and $\{g_1, \ldots, g_q\} \subset L^2(\mathcal{N})$, we denote their spectral representations by coefficients $\mathsf{A} = (a_{ij})_{k_1 \times q}$, where $a_{ij} = \langle \phi_i^{\mathcal{M}}, f_j \rangle_{\mathcal{M}}$, and $\mathsf{B} = (b_{ij})_{k_2 \times q}$, where $b_{ij} = \langle \phi_i^{\mathcal{N}}, g_j \rangle_{\mathcal{N}}$. The matrix $\mathsf{C}$ can be obtained by solving the following quadratic minimization problem:

$$\mathsf{C} = \mathrm{argmin}_{\mathsf{C} \in \mathbb{R}^{k_2 \times k_1}} \|\mathsf{C} \cdot \mathsf{A} - \mathsf{B}\|_{\mathrm{F}}^2 + \alpha \|\mathsf{C} \cdot \delta_{\mathcal{M}} - \delta_{\mathcal{N}} \cdot \mathsf{C}\|_{\mathrm{F}}^2 + \beta \sum_{i=1}^{q} \|\mathsf{C} \cdot \Lambda_{\mathcal{M}}^i - \Lambda_{\mathcal{N}}^i \cdot \mathsf{C}\|_{\mathrm{F}}^2, \quad \text{(E.1)}$$

where $\|\cdot\|_{\mathrm{F}}$ denotes the Frobenius norm, and the first term on the right hand side is the change of basis constraint. Two more regularization terms are introduced into the formula with tunable weights $\alpha, \beta > 0$. $\|\mathsf{C} \cdot \delta_{\mathcal{M}} - \delta_{\mathcal{N}} \cdot \mathsf{C}\|_{\mathrm{F}}$ enforces the isometry of the two manifolds where the matrices $\delta_{\mathcal{M}}$ and $\delta_{\mathcal{N}}$ are the spectral representation of the LB operators. The matrices $\Lambda_{\mathcal{M}}^i$ and $\Lambda_{\mathcal{N}}^i$ are called the orientation operator (Ren et al., 2018) and are defined by the functions $f_i$ and $g_i$, respectively. The commutator with the orientation operator incorporates extrinsic properties into the formulation and enforces the orientation of functional maps (i.e., which side of the 2D surface is pointing "outward").

Once the functional mapping (i.e., the $\mathsf{C}$ matrix) is recovered, point-to-point correspondence $T$ between manifold $\mathcal{M}$ and $\mathcal{N}$ can be obtained by mapping indicator functions of vertices on $\mathcal{M}$ to the corresponding functions on $\mathcal{N}$ because $T_F \delta_m = \delta_{T(m)}$ for any vertex $m \in \mathcal{M}$ (Ovsjanikov et al., 2012).

# F  DETAILS ON QM9 PROPERTY REGRESSION

**Dataset**  For the QM9 molecular property regression task, we align our input data with EGNN (Satorras et al., 2021) (with a total of 130,831 instances). 181 molecules failed during molecular surface extraction (`MSMS` failed to generate reasonable surface mesh for some molecules). We follow the same data split as EGNN, leading to 130,650 instances (99,862 for training, 17,719 for validation, and 13,069 for test). Only the atomic number and atomic positions are used as the initial molecular information, no extra handcrafted features are involved. We compute the molecular surface as described in Appendix C, with an average of 439 vertices and 47 eigenfunctions per molecular surface.

**Model Architecture and Performance**  Different from other graph-based models, we do not explicitly encode inter-atomic distance or chemical bonding information. Instead, we feed the molecular surface manifold as the model input, which contains local chemical information (Appendix C). Surface geometric features (i.e., curvature and the Heat Kernel Signatures) are not included, since we find these features have no contribution to predicting molecular properties. We stack 6 layers of HMR, followed by a global average pooling layer to aggregate information from all surface points to make a final property prediction. Batch normalization is applied to all MLPs. The mean absolute prediction error of all 12 properties are shown in Table F.1, results averaged over three independent runs with different random seeds.

Table F.1: Model performance on the QM9 dataset, reporting the Mean Absolute Error (MAE).

| Task | $\alpha$ | $\Delta \varepsilon$ | $\varepsilon_{\text{HOMO}}$ | $\varepsilon_{\text{LUMO}}$ | $\mu$ | $C_\nu$ | $G$ | $H$ | $r^2$ | $U$ | $U_0$ | ZPVE |
|---|---|---|---|---|---|---|---|---|---|---|---|---|
| Units | bohr$^3$ | meV | meV | meV | D | cal/(mol K) | meV | meV | bohr$^2$ | meV | meV | meV |
| NMP | .092 | 69 | 43 | 38 | .030 | .040 | 19 | 17 | .180 | 20 | 20 | 1.50 |
| SchNet | .235 | 63 | 41 | 34 | .033 | .033 | 14 | 14 | .073 | 19 | 14 | 1.70 |
| Cormorant | .085 | 61 | 34 | 38 | .038 | .026 | 20 | 21 | .961 | 21 | 22 | 2.02 |
| SE(3)-Tr. | .142 | 53 | 35 | 33 | .051 | .054 | – | – | – | – | – | – |
| EGNN | .071 | 48 | 29 | 25 | .029 | .031 | 12 | 12 | .106 | 12 | 12 | 1.55 |
| SEGNN | .060 | 42 | 24 | 21 | .023 | .031 | 15 | 16 | .660 | 13 | 15 | 1.62 |
| **HMR** | .102 | 59 | 37 | 31 | .037 | .040 | 40 | 41 | .653 | 44 | 42 | 2.97 |

**Resource Consumption** Computing the molecular surface and its Laplace-Beltrami eigenfunctions is computationally intensive. We performed data preprocessing in parallel on 64 CPUs, where molecular surface computation took 198 seconds, surface mesh refinement took 59 minutes, and solving eigenfunctions took 4.5 hours. With an average of 439 mesh vertices and 47 eigenfunctions, the entire QM9 dataset consumes 12.1 GB disk space (approximately 100 kB per molecule). The model for production contains 356,609 learnable parameters, most of which are linear transformation coefficients in MLPs. However, manifold harmonic message passing involves matrix multiplications bringing features back-and-forth between the real space and the generalized Fourier space, which is performed in a serial manner (instead of batch processing). This is because the number of surface vertices and Laplace-Beltrami eigenfunctions are different across different molecules. Therefore, we only perform batch operations on the MLPs, but not on message passing layers. We trained our model on NVIDIA A100 GPUs with 80 GB memory with a batch size of 32, which on average takes 240 seconds to train a single epoch (99,862 molecules), and 16 seconds for inference on the test set (13,069 molecules). For each molecule we keep its $N \times n$ feature matrix F ($N$ vertices and $n$ features), and the $N \times k$ eigenvectors (Z matrix) and its inverse matrix ($Z^\top B$ matrix, see Appendix B), which does not consume much GPU memory.

## G  DETAILS ON LIGAND-BINDING POCKET CLASSIFICATION

**Dataset** The dataset is obtained from Gainza et al. (2020) with 1,438 non-redundant protein structures that each corresponds to a list of bound ligands with their atom coordinates. We first generate protein surface meshes as described in Appendix C. To extract ligand binding pockets, we identify pocket vertices on the surface mesh that are within 4 Å to any atom of the ligand and extract the largest connected component of pocket vertices as the binding pocket. Binding pockets that contain $< 100$ vertices are removed and the LB eigenfunctions for the remaining pocket surfaces are calculated. 3 protein complexes failed in protein surfaces generation and 63 protein complexes failed in binding pocket extraction due to disconnected surface or too few vertices after surface refinement. In total, 2,254 binding pockets are extracted that bind to one of seven ligands: ADP (634), HEM (396), FAD (338), COA (263), NAD (232), SAM (222), and NAP (169). On average, 1,199 vertices and 190 eigenfunctions per pocket surface are obtained. The same training, validation, and test split as Gainza et al. (2020) are used and we obtained 1,634 training pockets (in 986 protein complexes), 202 validation pockets (in 112 protein complexes), and 418 test pockets (in 274 protein complexes). Similar to Gainza et al. (2020), we provide chemical information (hydrophobicity score, partial charge) as well as local geometric information (mean/Gaussian curvature and the Heat Kernel Signatures) as input features.

**Model Architecture and Performance** The HMR-based classification model contains 6 layers of HMR propagation layers followed by a global average pooling to aggregate information from all surface points of a pocket. A simple 2-layer MLP is used to classify pockets into seven ligand-binding classes. HMR is trained to minimize the cross-entropy loss for 400 epochs (approx. 20,000 iterations with a batch size of 32) and the one with the best balanced accuracy score on the validation set is selected. To compare with MaSIF-ligand, we calculated the balanced accuracy for multi-class classification (Fig. 6). Per class performance of our full model (geometric + chemical features) is shown as a confusion matrix (Fig. G.1).

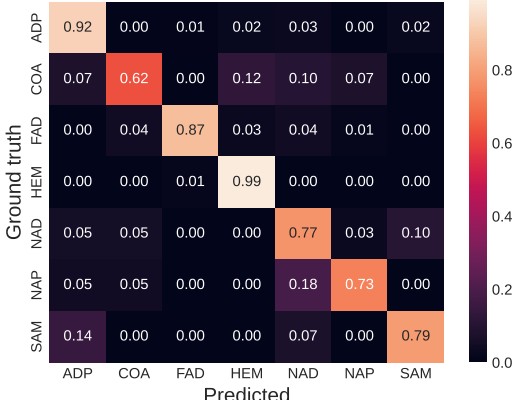

Figure G.1: Confusion matrix of ligand specificity predicted by HMR on the test set.

**Resource Consumption** Similar to QM9, we performed data preprocessing in parallel on 64 CPUs. For all 2,254 protein pocket cases: the protein surface computation took 306 seconds, surface pocket extraction and refinement took 330 seconds, and solving the eigenfunctions for the extracted pockets took 26 minutes. The processed dataset (N=2,254) takes 2.4 GB disk space (about 1.1 MB per protein pocket). The full model contains 407,047 trainable parameters. We trained our model on NVIDIA A100 GPUs with 80 GB memory with a batch size of 32, which on average takes 33 seconds to train a single epoch (1,634 protein pockets), and 9 seconds for inference on the test set (418 protein pockets). The model structure is almost identical to QM9 that shares the same batch processing design and has similar GPU memory consumption.

## H    DETAILS ON RIGID PROTEIN DOCKING

### H.1    DATASET

The original DIPS dataset (Townshend et al., 2019) contains interacting protein chains extracted from experimentally determined complex structures in the RCSB PDB database (https://www. rcsb.org/). While DIPS is designed to represent the interactions in protein complexes including homomeric proteins (i.e., protein complexes composed of identical proteins), we are particularly interested in interactions between different proteins in biological processes (e.g., antibody-antigen, enzyme-inhibitor, enzyme-substrates, etc.). To better resemble such interactions, we apply a modified pipeline based on DIPS, referred as DIPS-Het.

Specifically, we include the PDB entries in DIPS as well as newly deposited entries from the RCSB PDB database that (1) are determined using diffraction-based methods or electron microscopy, (2) have $< 3.5$ Å resolution, (3) are not hybrid protein complexes (e.g., protein-DNA complexes are excluded), and (4) have $< 5$ protein chains. Different from the original DIPS pipeline, we a) further remove the PDB entries with only homomeric interfaces (i.e., interactions between identical proteins), and b) adopt a one-vs-rest strategy that uses each one of the protein chains in the complex as the "ligand protein" and the rest protein chains as the "receptor protein" to form ligand-receptor pairs.

Similar to Ganea et al. (2021), we remove the cases with any protein chain sharing the same protein sequence clusters (at 30% sequence similarity) with DB5.5. Finally, we cluster proteins based on protein chain sequence similarity and separate these clusters into the training and validation set. This sequence-based split helps selecting a model with better generalizability. After removing proteins failed in surface mesh generation, DIPS-Het contains 11,373 training cases and 508 validation cases.

### H.2    EVALUATION METRICS

We evaluate the rigid protein docking results using four metrics: Complex RMSD, Interface RMSD, DockQ, and Success rate. Complex RMSD and Interface RMSD are used in Ganea et al. (2021): let

$\mathbf{Z}^* \in \mathbb{R}^{3 \times (n+m)}$ and $\mathbf{Z} \in \mathbb{R}^{3 \times (n+m)}$ be the $\alpha$-carbon coordinates of the ground truth and predicted protein complexes, respectively, were $m$ and $n$ are the number of $\alpha$-carbons in the receptor and ligand protein. After superimposing the complex structures using the Kabsch algorithm, Complex RMSD is calculated as $\sqrt{\frac{1}{n+m}||\mathbf{Z}^* - \mathbf{Z}||_F^2}$. Similarly, Interface RMSD is calculated using the same procedure but with the $\alpha$-carbon coordinates of interface residues ($< 8$ Å to the other protein's residues). Smaller RMSD value means the predicted structure is closer the ground truth structure.

In addition, we evaluate the overall quality of docking using DockQ (Basu & Wallner, 2016) and the success rate of achieving "Acceptable" or higher according to Méndez et al. (2003; 2005). Both metrics are based on three standardized criteria used by Critical Assessment of PRedicted Interactions (CAPRI): $L_{\mathrm{rms}}$ is the ligand (the smaller protein) RMSD calculated based on backbone atoms, after superimposing receptor's backbone atoms; $I_{\mathrm{rms}}$ is the backbone RMSD of interface residues, after superimposing the interface residues (residues with any atom is $< 10$ Å to atoms in the other protein); and $f_{\mathrm{nat}}$ is the recall in recovering residue-residue contacts between the proteins, where two residues are "in contact" if any pair of atoms from two residues has distance $< 5$ Å. DockQ is a continuous score between 0 and 1 (the higher the better), derived from $L_{\mathrm{rms}}$, $I_{\mathrm{rms}}$, and $f_{\mathrm{nat}}$. A docking result is considered as a "success" if it is ranked "Acceptable" or higher according to CAPRI's criteria, that is

$$f_{\mathrm{nat}} \geq 0.1 \wedge (L_{\mathrm{rms}} \leq 10.0 \vee I_{\mathrm{rms}} \leq 4.0)$$
$$\text{OR}$$
$$f_{\mathrm{nat}} \geq 0.3$$

Both DockQ score and CAPRI ranking is calculated using the DockQ package (`https://github.com/bjornwallner/DockQ/`).

## H.3 RESOURCE CONSUMPTION

**Dataset**   We report the rigid protein docking resource consumption using the Docking Benchmark 5.5 (DB5.5) dataset (Guest et al., 2021), which contains 253 pairs of representative protein complex structures. This is the gold-standard test set to evaluate protein docking model performance.

We preprocessed the DB5.5 dataset in parallel on 64 CPUs. Adding hydrogen atoms to the PDB data took 34 seconds (using `reduce`), computing the solvent-excluded molecular surface with `MSMS` took 6 seconds, triangle mesh refinement by `PyMesh` took 3.4 minutes, and computing the Laplace-Beltrami eigenfunctions took 18 minutes with `scipy` (`eigsh`)[3]. The average number of vertices per protein (i.e., ligand or receptor) is 3,380, with 215 calculated Laplace-Beltrami eigenfunctions (maximum eigenvalue capped to 0.3, which is determined empirically). The total storage space for the processed dataset is 2 GB (about 8 MB per protein, much larger than QM9 molecules). For our training set with 11,781 protein complexes, the storage space is 101 GB.

**Inference**   The HMR inference time in predicting DB5.5 proteins complexes is shown in Table H.1, in comparison to EQUIDOCK, HDOCK, and ATTRACT. For HMR, data preprocessing took 80% of inference time, functional maps (docking pose prediction) took 19%, while binding site prediction took only 1% of time.

Table H.1: Rigid protein docking inference time on DB5.5 dataset (per protein complex docking time averaged over 253 cases).

| Method | Time |
|---|---|
| HDock | 884.9 sec |
| ATTRACT | 882.7 sec |
| EquiDock | 3.9 sec |
| **HMR** (Top 1) | 5.8 sec |
| **HMR** (Top 3) | 6.6 sec |

---

[3]Solving the generalized eigenvalue problem should be done with less parallel processes to allocate more CPU resources to each solver for better efficiency.

## H.4 Distributions of learned propagation parameter

We visualize the distribution of propagation parameters: propagation time $t$, frequency filter mean $\mu$ and variance $\sigma$, learned from data. The histogram of selected layers are shown in Fig. H.1.

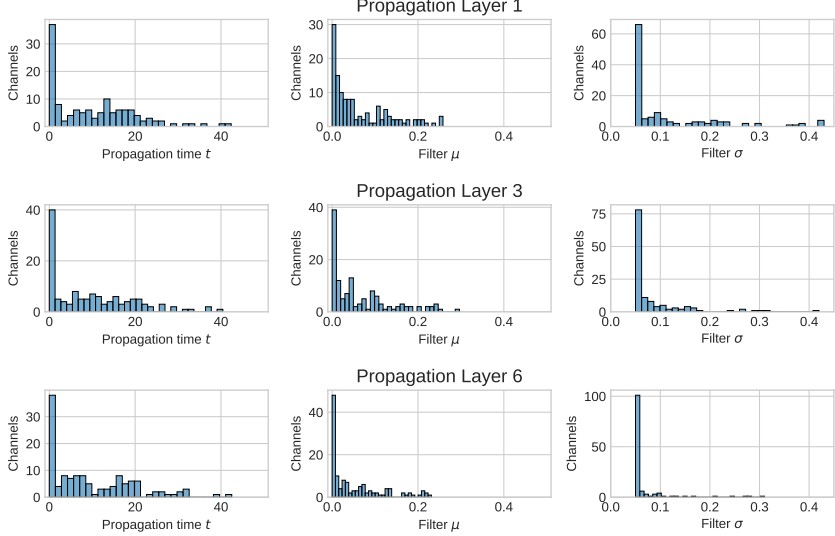

Figure H.1: Distribution of learned propagation time $t$ and frequency filter ($\mu$ and $\sigma$) at selected propagation layers.

## H.5 Analysis of Different Message Passing Mechanisms

We study the effectiveness of proposed Harmonic Message Passing mechanism by comparing it with spatial graph message passing, a "Graph Message Passing" model where Harmonic Massage Passing is replaced by Graph Attention Network (Veličković et al., 2018), and with a "No Message Passing" model where only a multilayer perceptron (MLP) is used to encode per-vertex features. Experimental results show that the Harmonic Message Passing model outperforms Graph Message Passing and No Message Passing models, suggesting its higher efficiency in propagating information on protein surface meshes.

Table H.2: Rigid Protein Docking Performance with Different Message Passing Models

|  | Complex RMSD | | Interface RMSD | | DockQ | |
|---|---|---|---|---|---|---|
| Model | Median | Mean | Median | Mean | Median | Mean |
| **Harmonic Message Passing** | **12.01** | **12.09** | **10.90** | **11.43** | **0.06** | **0.22** |
| Graph Message Passing | 15.41 | 15.67 | 13.38 | 14.37 | 0.03 | 0.08 |
| No Message Passing | 16.44 | 17.23 | 14.11 | 16.11 | 0.02 | 0.04 |

## H.6 Ablation Studies on Input Features

To assess the importance of input chemical and geometric features for rigid protein docking, we compare the performance of HMR ("Full model") to models trained with only chemical or geo-metric features: the "Chem only" model only includes chemical features (i.e., atom types, residue types, residue's hydrophobicity, and whether the atom is an $\alpha$-carbon); the "Geom only" model only contains features related to manifold geometry, including Gaussian curvature, mean curvature, and Heat Kernel Signatures. Table H.3 shows that the model performance drops upon removing either chemical or geometric features. And the chemical features is more critical for HMR in the rigid protein docking task.

Table H.3: Ablation studies on input features

| Model | Complex RMSD | | Interface RMSD | | DockQ | |
|---|---|---|---|---|---|---|
| | Median | Mean | Median | Mean | Median | Mean |
| **Full model** | **12.01** | **12.09** | **10.90** | **11.43** | **0.06** | **0.22** |
| Chem only | 13.96 | 13.95 | 12.73 | 13.33 | 0.04 | 0.17 |
| Geom only | 17.87 | 19.15 | 16.58 | 18.46 | 0.01 | 0.03 |

## H.7 CROSS EXPERIMENTS WITH EQUIDOCK

Compared to EQUIDOCK, HMR differs both in the model and the dataset used for training. To understand which aspect contributes more to the performance of HMR, we conduct cross experiments to train HMR on the original DIPS (used in EQUIDOCK) and EQUIDOCK trained on DIPS-Het (proposed by us, used in HMR). The same training and validation sets are used, and all models are evaluated on DB5.5. As summarized in Table H.4, model structure and training dataset are both important: HMR achieved higher performance than EQUIDOCK when both models are trained on DIPS dataset; the performance of HMR further improves when it is trained on DIPS-Het dataset.

Table H.4: Compare EQUIDOCK and HMR trained on DIPS or DIPS-Het datasets

| Model | Training data | Complex RMSD | | Interface RMSD | | DockQ | |
|---|---|---|---|---|---|---|---|
| | | Median | Mean | Median | Mean | Median | Mean |
| HMR | DIPS | 15.68 | 16.04 | 14.81 | 15.38 | 0.02 | 0.07 |
| | DIPS-Het | 12.01 | 12.09 | 10.90 | 11.43 | 0.06 | 0.22 |
| EQUIDOCK | DIPS | 17.27 | 18.09 | 15.04 | 16.11 | 0.02 | 0.03 |
| | DIPS-Het | 17.22 | 17.65 | 14.27 | 14.76 | 0.02 | 0.04 |

## H.8 DB5.5 TEST CASES FROM EQUIDOCK

For a fair comparison, we further compare HMR with EQUIDOCK on the 25 cases in DB5.5 selected as a test set in Ganea et al. (2021). We report the EQUIDOCK model fine-tuned on DB5.5, as originally reported in Ganea et al. (2021). As shown in Table H.5, HMR still outperforms EQUIDOCK, despite not being fine-tuned on DB5.5.

Table H.5: Rigid docking performance on DB5.5 test set

| Model | Complex RMSD | | Interface RMSD | | DockQ | | Success rate |
|---|---|---|---|---|---|---|---|
| | Median | Mean | Median | Mean | Median | Mean | ($\geq$ Acceptable) |
| HMR (Top 1) | 13.10 | 12.42 | 11.05 | 11.68 | 0.07 | 0.15 | 0.20 |
| HMR (Top 3) | 11.35 | 11.18 | 10.02 | 10.03 | 0.08 | 0.16 | 0.20 |
| EQUIDOCK | 14.13 | 14.72 | 11.97 | 13.23 | 0.04 | 0.05 | 0.00 |

## H.9 MODEL PERFORMANCE AT DIFFERENT RESOLUTIONS

We perform the rigid docking experiment using HMR at different resolutions. Specifically, we cap the largest eigenvalue of the ligand and receptor protein surface manifolds used in the model. Since eigenfunctions with larger eigenvalues exhibit higher spatial resolutions, restricting the eigenvalues effectively constrains the model resolution. The results are shown in Table H.6. In general, we observe worse model performance with fewer eigenfunctions for harmonic message passing and functional maps. Interestingly, the metrics of the binding site prediction module (i.e., AUC and AP)

are less sensitive to resolution than the rigid docking module (powered by functional maps). This means the model could still infer where the binding site is with lower resolution, yet the quality of learned functional correspondence decreases, leading to worse docking power.

Table H.6: HMR rigid protein docking performance at different spatial resolutions (quantified by the largest eigenvalue of surface manifold). AUC (area under the receiver operating characteristic curve) and AP (average precision) are metrics of the binding site prediction module.

| max eigenvalue | AUC | AP | CRMSD (Top 1) | CRMSD (Top 3) |
|---|---|---|---|---|
| 0.30 | 0.84 | 0.54 | 12.11 | 8.36 |
| 0.25 | 0.84 | 0.53 | 13.14 | 8.9 |
| 0.20 | 0.84 | 0.54 | 11.91 | 9.53 |
| 0.15 | 0.83 | 0.53 | 12.19 | 9.91 |
| 0.10 | 0.82 | 0.52 | 12.90 | 10.02 |
| 0.05 | 0.81 | 0.51 | 13.84 | 11.32 |

## H.10  TRAINING DETAILS

Table H.7: Hyperparameter choices of HMR and the training phase settings

| Hyperparameters | Values |
|---|---|
| **Binding Site Prediction Module** | |
| Number of Feature Propagation Blocks | 2 |
| Number of HMR Layers | 3 |
| Dimension of Propagated Features $d_k$ | 128 |
| Number of Attention Heads | 4 |
| Dropout Rate | 0.1 |
| NCELoss Temperature | 10 |
| NCELoss Number of Point Samples | 50 |
| NCELoss Weight $\lambda$ | 0.1 |
| **Training** | |
| Batch Size | 4 |
| Epoch | 80 |
| Learning Rate | $5\times10^{-4}$ |
| Optimizer | Adam |
| Learning Rate Scheduler | Cosine Annealing |

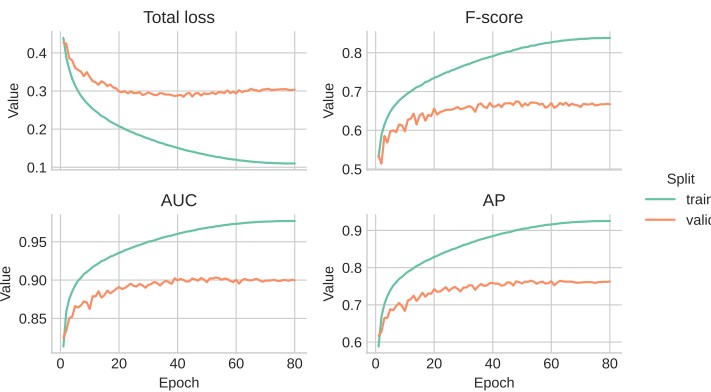

Figure H.2: HMR learning curve for the rigid protein docking task. Model with the best validation average precision (AP) score in binding site prediction is selected for testing on DB5.5.

