# OpenReview forum: "Learning Harmonic Molecular Representations on Riemannian Manifold"
_ICLR.cc/2023/Conference — ICLR 2023 poster_

### Official Review · Reviewer_R5jV · 2022-10-20

**Confidence:** 2
**Correctness:** 3
**Technical Novelty And Significance:** 4
**Empirical Novelty And Significance:** 3
**Recommendation:** 8

**Clarity, Quality, Novelty And Reproducibility:**

Clarity: While I was able to understand this paper (despite being new to LB eigenfunctions), I think the clarity is suboptimal (see weakness).

Quality: I found it convincing for the method to be potentially impactful for molecular representation learning. However, empirical evaluation is could be improved to enough to support this fact.

Novelty: I am aware of existing works on spectral graph convolution between point cloud, graph, and mesh representation of molecules/proteins. However, The idea of harmonic message passing between LB eigenfunctions is very new to me. If this is truly novel, I believe the novelty of this work is very strong.


**Strength And Weaknesses:**

Strength.

This paper proposes a novel molecular representation learning framework inspired by molecular surface representation and harmonic analysis. To my knowledge, various geometric deep learning works have focused on message passing between mesh vertices rather than the surface regions (LB eigenfunctions). If my knowledge is correct, I believe this work to be quite significant.

I also appreciate how the experiments are done on both molecules & proteins with various tasks (molecular property prediction, binding site prediction, rigid docking simulation). This makes the empirical benefit of the proposed method convincing.

The proposed method looks quite scalable, so I think this method might be potentially useful in large-scale scenario where computation time is important.

Weakness.

In the experiments, it is not clear whether if the complexity of preprocessing molecular surface is considered. I find this confusing since the EquiDock authors mentioned how they avoided molecular surface preprocessing due to high complexity.

The experimental results for molecular property prediction is not exactly state-of-the-art, but I think this is fine given novelty of the work. Nevertheless, I suggest comparing with more recent works like "Geometric and physical quantities improve e(3) equivariant message passing, ICLR 2022" for molecular property prediction (QM9) for completeness of the paper.

I think clarity the description of harmonic message passing could be improved. For example, the input and the output of harmonic message passing is expressed as a function. However, to my understanding, the features being actually updated correspond to different coefficients for the LB eigenfunctions. I think the matrix representation of the function operator (Fig 3) should be described in the text. Also, the symbols in Fig 3 has no corresponding explanation, e.g., what is $\Phi$?

The authors claim this method to be scalable to large scale molecules and proteins. However, although the logic is convincing, the authors do not design any experiements to support this fact. When HMP is compared to Equidock in term of preprocessing + inference time, is it faster?

(minor) I find that the experimental setting differs from Equidock. Could you explain the reasoning behind this?




**Summary Of The Paper:**

This paper proposes a framework to learn Riemannian manifold representation using harmonic message passing. To this end, the authors (1) pre-process molecules (or macromolecules like proteins) as a surface function using triangulation, (2) decompose the surface function into components with LB eigenfunctions and coefficients, and (3) perform spectral message passing between the component-wise features. Importantly, the proposed harmonic message passing is motivated by the interpretation of heat diffusion as a message passing between surface regions (LB eigenfunctions and coefficient) with different temperatures (eigenvalues). The proposed methodology is applied for molecular property prediction, binding site prediction, and rigid docking simulation. The empirical results are promising and competitive with respect to the recently proposed molecular representation learning algorithms.

**Summary Of The Review:**

This paper proposes a very novel idea for molecular representation learning. While the experiments could be slightly improved, I think this paper is potentially very impactful and interesting for molecular representation learning community

---

> ### Author Response · Authors · 2022-11-14
> **Response to Reviewer R5jV - Part 1**
>
> **Comment 1**: In the experiments, it is not clear whether the complexity of preprocessing molecular surface is considered. I find this confusing since the EquiDock authors mentioned how they avoided molecular surface preprocessing due to high complexity.
>
> **Response 1**: We thank the reviewer for pointing this out and we added complexity analyses of our model, including preprocessing time, storage size, and inference time. Briefly, EquiDock is more efficient than molecular surface-based methods due to the lack of molecular surface preprocessing, but in exchange of the representation power and performance of the model. Overall, HMR has a comparable protein docking inference time to EquiDock (5.8 sec vs. 3.9 sec per protein complex, including preprocessing time). Please refer to the General Response and Appendix F-H of the revised manuscript for details.
>
> On the other hand, our preprocessing time is dominated by the computation of Laplace-Beltrami eigenfunctions through solving eigenvalue problems for elliptic partial differential equations (PDEs). While classical eigensolvers have complexity of O(N$^3$) for a surface with n vertices, significantly more efficient deep learning solvers for PDEs have emerged in recent years [1-3] that might further reduce the complexity in solving the Laplace eigenfunctions. Application of the neural network eigensolvers in our setting is far out of the scope of the current article, and we leave it for future study.
>
> [1] Raissi et al., J. Comput. Phys. 378 (2019): 686-707.
>
> [2] Lu et al. arXiv preprint arXiv:1910.03193.
>
> [3] Li et al. J. Comput. Phys. 453 (2022): 110939.
>
> **Comment 2**: I suggest comparing with more recent works like SEGNN for molecular property prediction (QM9) for completeness of the paper.
>
> **Response 2**: We thank the reviewer for the recommendation and have added a comparison with SEGNN [4] to Table 1 and Table F.1. We include the QM9 task mainly to showcase HMR's versatility to encode molecules at different scales (small molecules vs large biomolecules in protein rigid docking problem) with similar performance to current state-of-the-art GNN-based models. We expect GNN-based methods to have stronger performance given their ability to aggregate local information and the small size of molecules in QM9 dataset; in comparison, our model excels in larger molecules as we demonstrated in the rigid docking task. We added a paragraph in Conclusions and Outlook section to further clarify the advantages and limitations of HMR.
>
> [4] Brandstetter et al., arXiv preprint arXiv:2110.02905.
>
> **Comment 3**: I think clarity in the description of harmonic message passing could be improved. For example, the input and the output of harmonic message passing is expressed as a function. However, to my understanding, the features being actually updated correspond to different coefficients for the LB eigenfunctions. I think the matrix representation of the function operator (Fig 3) should be described in the text. Also, the symbols in Fig 3 has no corresponding explanation, e.g., what is $\Phi$?
>
> **Response 3**: We thank the reviewer for pointing this out. We have updated the caption of Fig 3 to clarify the notations in the figure. $\Phi$ is the matrix of column-wise stacked eigenfunctions.
> Regarding the relationship between a surface function and the corresponding LB coefficients -- a function can be analytically decomposed into the linear combination of the LB eigenfunctions, each eigenfunction $\phi_i$ has a coefficient $c_i$. Therefore, updating the coefficients is equivalent to updating the surface function, and harmonic message passing is one particular method to update the coefficients. The output of the harmonic message passing block are the updated coefficients in spectral domain, which can be transformed into real space as surface functions (through an inverse Fourier transform). In other words, the per-vertex features (in real space) and the coefficients (in spectral domain) are equivalent descriptions of the surface function. We provide more detailed explanations in Appendix A,B for your reference.

---

> > ### Author Response · Authors · 2022-11-14
> > **Response to Reviewer R5jV - Part 2**
> >
> > **Comment 4**: The authors claim this method to be scalable to large scale molecules and proteins. However, although the logic is convincing, the authors do not design any experiments to support this fact. When HMP is compared to Equidock in term of preprocessing + inference time, is it faster?
> >
> > **Response 4**:
> > 1. We demonstrate the scalability of HMR using three experiments on molecular systems of various sizes: QM9 small molecules, protein-ligand binding pocket, and large proteins. The unique advantage of HMR is that the Shape-DNA is size-invariant. With larger molecules, we need to compute more surface vertices and Laplace-Beltrami eigenfunctions, but the underlying learning mechanism remains identical to that for small molecules. In fact, we think the HMR framework is better suited for modeling large biomolecules rather than small molecules, as supported by our experimental results on the QM9 and rigid protein docking tasks.
> > 2. We have added detailed runtime of our HMR model for all three tasks. The surface preparation step takes a significant amount of time (e.g., 80% of total inference time for the rigid protein docking task). But overall HMR has a comparable protein docking inference time to EquiDock (5.8 sec vs. 3.9 sec per protein complex, including preprocessing time), both are two orders of magnitude faster than ATTRACT (882.7 sec) and HDOCK (884.9 sec). Please refer to the General Response and Appendix F-H in the revised manuscript for more details.
> >
> > **Comment 5**: Importantly, the proposed harmonic message passing is motivated by the interpretation of heat diffusion as a message passing between surface regions (LB eigenfunctions and coefficient) with different temperatures (eigenvalues).
> >
> > **Response 5**: We thank the reviewer for acknowledging the novelty of our work. In addition, we would like to point out that eigenvalues are associated with the surface Dirichlet energy, but not with surface temperature (see Appendix A for details). The temperature can be viewed as a surface function, which can be decomposed into the linear combination of LB eigenfunctions. We want to make this clarification just in case of misunderstanding.
> >
> > **Comment 6**: I find that the experimental setting differs from Equidock. Could you explain the reasoning behind this?
> >
> > **Response 6**: Thanks for bringing this up. The major difference between our experimental setting and EquiDock's is that EquiDock separately evaluates the model performance on a test set of DIPS and a test set of DB5.5 after fine-tuning (DB5.5 contains a total of 253 protein complexes, which is further split into 203/25/25 for fine-tuning). We use a more challenging setting that trains our model only on DIPS/DIPS-Het and tests on the complete DB5.5 dataset with 253 protein complexes.
> > We chose this setting based on following reasons:
> > i) Evaluation on the entire DB5.5 is more informative than on a subset of DIPS, as DB5.5 is a high-quality dataset manually curated by experts and can better reflect models' performances on predicting docking poses of real-world interest (e.g., enzyme-substrate, antibody-antigen).
> > ii) DB5.5 is a non-redundant dataset where each case represents a distinct type of protein-protein interaction. We did not further split DB5.5 for finetuning due to its already small size, and testing on the complete DB5.5 dataset (N=253) rather than a subset of it (N=25) provides a more comprehensive and robust evaluation.
> > iii) Our setting challenges the model and can reveal the generalizability of a model when testing on a different dataset.
> >
> > Nevertheless, we also provided a direct comparison with EquiDock (finetuned on DB5.5) by evaluating our model on the 25 DB5.5 test cases (Table H.5). Despite not fine-tuned on DB5.5, HMR still presents better performance than EquiDock.
> >
> > Lastly, we would like to point out that having a standard test set (e.g., DB5.5) with well-defined and reasonable evaluation metrics is significant for drug discovery research. For the docking task, RMSD alone is not very informative, so we also report the DockQ and success rate in our experiments. The research community should collaboratively work on standardizing more benchmark datasets for important drug discovery tasks (e.g., protein-protein interactions).

---

> > > ### Comment · Reviewer_R5jV · 2022-11-17
> > > **Thank you for the detailed response!**
> > >
> > > I have read the author's thoughtful response and other reviews. Although I am more familiar with the 3D GNN architectures (and not familiar with the surface DNNs), I think this is a strong paper that brings a nice & novel perspective of the molecular representation problem.

---

> ### Author Response · Authors · 2022-11-16
> **We look forward to hearing your feedback!**
>
> Dear reviewer,
>
> Thanks for taking your time to share your feedback.
> We revised the manuscript and performed further analysis to address your concerns.
> Please see our inline response for more details.
> We look forward to hearing more about your thoughts, and would be happy to answer more follow-up questions.

---

### Official Review · Reviewer_tNJq · 2022-10-24

**Confidence:** 4
**Correctness:** 3
**Technical Novelty And Significance:** 3
**Empirical Novelty And Significance:** 3
**Recommendation:** 6

**Clarity, Quality, Novelty And Reproducibility:**

The paper is well-written in general. However, some figures are not clear. For instance, the notation given in the figures should be defined clearly in the captions or the figures should be given as the notations are defined.

In the analyses, the proposed method outperforms various state-of-the-art. Could you please also elaborate dynamics of networks by analyzing learning curves obtained during training? Do you also follow a particular recipe while designing network architectures? An ablation study of different networks and optimizers can highlight network dynamics as well.


**Strength And Weaknesses:**

The paper addresses an important problem of encoding 3D molecular structures in deep neural networks. The paper is well written and the proposed methods are examined in various tasks.

Some parts of the presentation can be improved and additional results can be provided as suggested below.

**Summary Of The Paper:**

This paper proposes propose a Harmonic Molecular Representation learning (HMR) framework,which offers a multi-resolution representation of molecular geometric and chemical properties on 2D Riemannian manifold. In order to realize efficient spectral message passing over the surface manifold for better molecular encoding, a harmonic message passing method is proposed. The proposed method outperforms the state-of-the-art deep learning models for the rigid protein docking challenge, and shows comparable predictive power to current models in small molecule property prediction tasks.

**Summary Of The Review:**

The paper is well written in general, and the proposed method was examined in comparison with state-of-the-art in various experimental analyses.

The recommended updates can improve presentation of the paper and elucidate dynamics of the proposed learning methods and networks.

---

> ### Author Response · Authors · 2022-11-14
> **Response to Reviewer tNJq**
>
> **Comment 1**: The notation given in the figures should be defined clearly in the captions or the figures should be given as the notations are defined.
>
> **Response 1**: Thanks for pointing this out. In the revised manuscript, we updated the figure captions to clearly describe each plot. In addition, we realize that some concepts may not be straightforward to follow for readers unfamiliar with geometry processing techniques. To that end, we modified the descriptions in the main text to be more intuitive, while more rigorous and complete definitions and derivations have been moved to the Appendix. After all, we would like to share our ideas with the readers and hope to see more researchers working on solving relevant challenges.
>
> **Comment 2**: Could you please also elaborate dynamics of networks by analyzing learning curves obtained during training?
>
> **Response 2**: We included an example learning curve from our rigid protein docking task in Figure H.2 (page 25). The experiment was performed using Adam optimizer and cosine annealing scheduler with initial learning rate 5E-4 and we trained the model for 80 epochs. Model with the best validation average precision (AP) score in binding site prediction is selected for testing on DB5.5. We observed smooth training curves during the experiment: the learning process saturates roughly after 40 epochs and no obvious over-fitting is observed. We will add relevant discussions to the main text if later we have a flexible space limit.
>
> **Comment3**: Do you also follow a particular recipe while designing network architectures?
>
> **Response 3**: This is an interesting question. In the revised manuscript, we illustrated our methodology using some analogies. Specifically:
> 1. For designing the harmonic message passing network, we emulated the message passing framework in GNNs. However, since a manifold is continuous while graphs are discrete, explicitly modeling vertex-to-vertex message passing for a discretized manifold seems a bit awkward. Therefore, we learned from Joseph Fourier about harmonic analysis techniques to solve the problem.
> 2. For designing the rigid protein docking workflow, we made an analogy to solving a puzzle problem. The common-sense strategy for solving a puzzle is the recipe for designing the docking workflow: (1) locate the missing piece (binding site prediction), (2) complete the puzzle by matching the shape and pattern (rigid docking using functional maps). Similar strategies are widely used in the 3D computer vision research community for point cloud registration tasks [1].
> 3. Regarding the neural network architectures, we basically started from a simple binding site prediction model and updated each component in a greedy approach. We then implemented the functional maps module to complete the docking procedure. The docking pipeline is not end-to-end, which makes it look a bit complicated (it is indeed non-trivial), yet we were not able to develop a better approach to achieve the same performance.
>
> [1] Elbaz et al., CVPR (2017): 4631-4640.
>
> **Comment 4**: An ablation study of different networks and optimizers can highlight network dynamics as well.
>
> **Response 4**: We performed ablation studies in Appendix H6-9 (included in the original version), which demonstrates the advantage of our proposed framework from different aspects.
> 1. Generally speaking, we find the rigid protein docking task to be quite challenging. The lack of available protein complex structures (non-redundant) is a severe problem for deep learning-based models. This limits the model generalizability.
> 2. Empirically, we find the network to be non-sensitive to hyperparameters or optimizers (Adam outperforms SGD in our experiments).
> 3. In most cases where HMR failed to deliver a reasonable protein complex structure, the protein binding interface is typically not well-predicted. The functional maps module could predict structures close to the ground truth given proper binding interfaces.

---

> ### Author Response · Authors · 2022-11-16
> **We look forward to hearing your feedback!**
>
> Dear reviewer,
>
> Thanks for taking your time to share your feedback.
> We revised the manuscript and performed further analysis to address your concerns.
> Please see our inline response for more details.
> We look forward to hearing more about your thoughts, and would be happy to answer more follow-up questions.

---

### Official Review · Reviewer_GJXL · 2022-10-24

**Confidence:** 4
**Correctness:** 3
**Technical Novelty And Significance:** 3
**Empirical Novelty And Significance:** 3
**Recommendation:** 6

**Clarity, Quality, Novelty And Reproducibility:**

The paper is well-motivated and the technical part is clear and easy to follow. The experiments are not designed properly to evaluate the surface representation approach. I am not an expert in protein surface representation, but the approach is novel to the extent of the reviewer’s knowledge. The authors provide code in the supplementary material to reproduce their results.

**Strength And Weaknesses:**

Strengths:
- The Laplace-Beltrami operator as a harmonic filter provides a principled approach to encode interactions at different resolutions of molecular surfaces.
- The model outperforms EquiDock in the protein rigid docking task.

Weaknesses:
- The comparison with GNN baselines is not very informative. It is well-known that molecular surface representation captures very different information compared with GNNs. The HMR also uses surface geometry features and chemical features which are not present in the GNN baselines. It is not surprising that HMR can outperform the GNN based EquiDock baseline due to the additional information. I think comparing it with another protein-surface representation baseline like MaSIF [Gainza et al., Nature Methods 17.2 (2020): 184-192.] or Multi-Scale Protein Representation [Somnath et al., NeurIPS 2021] is needed to demonstrate the usefulness of this new molecular surface representation approach.
- In Table 2, the runtime of the HMR approach is not reported. The key advantage of EquiDock is its significantly faster inference compared conventional docking software. The surface construction might be slow for the HMR method. How does the inference time compare with EquiDock, Attract, and Hdock?


**Summary Of The Paper:**

This paper develops a harmonic representation of a molecular surface using the Laplace-Beltrami eigenfunctions. The approach is applied to predict the properties of small molecules and predict the docking pose of proteins.

**Summary Of The Review:**

In summary, this paper develops a novel harmonic representation of the molecular surface and applies the approach to two tasks: QM9 and protein rigid docking. Despite the potential improvement in the rigid docking task, the HMR approach may benefit from the additional geometric and chemical features in the surface representation. Comparing it to another surface representation baseline which encodes the same information is needed to demonstrate the advantage of the harmonic representation from the reviewer’s perspective.

---

> ### Author Response · Authors · 2022-11-14
> **Response to Reviewer GJXL**
>
> **Comment 1**: The comparison with GNN baselines is not very informative. The HMR also uses surface geometry features and chemical features which are not present in the GNN baselines. It is not surprising that HMR can outperform the GNN based EquiDock baseline due to the additional information.
>
> **Response 1**: Thanks for your feedback. Our understanding is that GNNs and HMR apply different feature extraction strategies to the exact same molecular information. Therefore, it is the features we use that are different from those in GNNs, but we are not utilizing additional information. Please see our justifications below:
> 1. We would like to clarify that all the information used in our model is derived from the same molecular structure data as used in other GNN baselines (e.g., QM9 and protein docking). Therefore, since we are using the exact same raw molecular information, the additional derived features are the advantages of the proposed representation learning method.
> 2. Our paper proposes a different way to represent/encode molecular information (instead of introducing new information), which could be a better alternative in some specific applications.
> 3. It is correct that GNNs do not include surface chemical or geometric features, yet likewise HMR does not utilize bonding or inter-atomic distance information as GNNs do. These different molecular representations exhibit their unique advantages in different tasks (e.g., GNN-based models are the state-of-the-art for QM9 tasks, while our surface-based model shows better performance in the protein docking task).
> 4. Furthermore, we think that introducing the molecular Shape-DNA is essentially introducing a strong inductive bias to the deep learning model, where features are constrained on the surface manifold and can be analytically propagated according to the Laplace-Beltrami eigenfunctions. This might explain the success of the proposed model in the protein docking task, where training data is limited. It is possible that when training data is abundant, such strong inductive bias may limit the model expressiveness. We are actively exploring this interesting topic.
>
> **Comment 2**: I think comparing it with another protein-surface representation baseline like MaSIF or Multi-Scale Protein Representation is needed to demonstrate the usefulness of this new molecular surface representation approach.
>
> **Response 2**: Thanks for your suggestion, we included an additional task to compare HMR with MaSIF [1] on a protein ligand-binding pocket classification task. In this experiment, HMR outperforms MaSIF-ligand on the protein pocket classification problem using a set of similar surface chemical and geometric features. We attribute the success of HMR in this task to the multi-resolution features which capture both the shape of the pocket and relevant chemical information, in contrast to the fixed-size geodesic CNN filters used in the MaSIF-ligand model. Please see Section 5.2 of the revised manuscript for details.
>
> [1] Gainza et al., Nature Methods 17.2 (2020): 184-192.
>
> **Comment 3**: The runtime of the HMR approach is not reported. The surface construction might be slow for the HMR method. How does the inference time compare with EquiDock, Attract, and Hdock?
>
> **Response 3**: Thanks for the suggestion and we have added detailed runtime of our HMR model for all three tasks. It is correct that the surface preparation step takes a significant amount of time (e.g., 80% of total inference time for the rigid protein docking task). But overall HMR has a comparable protein docking inference time to EquiDock (5.8 sec vs. 3.9 sec per protein complex, including preprocessing time), both are two orders of magnitude faster than ATTRACT (882.7 sec) and HDOCK (884.9 sec). Please refer to Appendix F-H in the revised manuscript for more details.

---

> > ### Comment · Reviewer_GJXL · 2022-11-18
> > **Thank you for your response**
> >
> > Thank you for your response! The new MaSIF baseline and runtime of HMR have addressed most of my concerns. I will update my evaluation.

---

> ### Author Response · Authors · 2022-11-16
> **We look forward to hearing your feedback!**
>
> Dear reviewer,
>
> Thanks for taking your time to share your feedback.
> We revised the manuscript and performed further analysis to address your concerns.
> Please see our inline response for more details.
> We look forward to hearing more about your thoughts, and would be happy to answer more follow-up questions.

---

### Official Review · Reviewer_tWAd · 2022-10-29

**Confidence:** 4
**Correctness:** 2
**Technical Novelty And Significance:** 3
**Empirical Novelty And Significance:** 3
**Recommendation:** 6

**Clarity, Quality, Novelty And Reproducibility:**

Clarity: the paper is overall well-written, and the descriptions of methods are clear.

Quality: Some statements regarding the motivation for using surface representations are not fully justified.

Novelty: the idea and methods are novel.

**Strength And Weaknesses:**


### Strengths

(+) The studied problem of the “shape-activity” relationship as an alternative to the “structure-activity” relationship is interesting and can provide some insights into understanding the system.

(+) The proposed method itself, including surface representation and information aggregation, is interesting and novel for geometric graph learning. I’m sure it can also be applied to many other scenarios than molecules.

(+) The proposed rigid protein docking framework has additional technical contributions.

### Weaknesses

(-) The use of surface only is not well-motivated nor justified. Although the authors claim that “molecular surface displays chemical and geometric patterns which fingerprint a protein’s mode of interactions with other biomolecules”, some analysis or empirical results are needed to justify the claim and make the proposed surface representation more convincing. Moreover, I agree that the surface may be important for binding problems to some degree, but the claim “this possibly explains the success of predicting these properties using only molecular surface information, since the surface is approximately an isosurface of the molecular electron density field” is not convincing enough and may need additional justifications.

(-) Current experimental results are not convincing enough to show the significance and effectiveness of the proposed methods. The experimental results on QM9 do not show a significant improvement to the baselines (some are worse). It indicates the practical application of the proposed method may be limited to only a few specific tasks. For the binding task, the EquiDock can be a relatively weak baseline. The authors may want to include stronger baselines such as EquiBind [1].

(-) There is a lack of details or comparisons on the method's computation complexity in terms of runtime and memory cost. Also, it would be better if the authors include results on how increasing the dimensions (surface resolution) can improve the performance and increase the computation cost.

[1] Stark et al. EquiBind: Geometric Deep Learning for Drug Binding Structure Prediction. ICML 2022.


**Summary Of The Paper:**

The authors study the molecule representation problem from a novel aspect based on the surface Riemannian manifold. Specifically, the authors adopt the Shape-DNA techniques to obtain surface representation and propose the information aggregation for the surface. The authors further design the frameworks for molecular property prediction and binding prediction.

**Summary Of The Review:**

Overall I like the idea in this paper. Using representation and message passing of surface is a pretty natural approach to the binding problem. The proposed idea is novel. However, there are concerns that some important claims are not fully justified and experimental results are not strong enough.

---

> ### Author Response · Authors · 2022-11-14
> **Response to Reviewer tWAd - Part 1**
>
> **Comment 1**: The use of surface only is not well-motivated nor justified. Some analysis or empirical results are needed to justify the claim and make the proposed surface representation more convincing.
>
> **Response 1**: Thanks for your suggestion. We realize that in the original manuscript, the sudden transition from NMP to the proposed HMR is a bit abrupt and not convincing for the readers. In the revised version, we first discuss the significance of molecular surface modeling in studying inter-molecular interactions before introducing the proposed HMR. In fact, the molecular surface has been widely used in biology research for decades [1,2]. Recently, some molecular surface-based deep learning models have been developed for drug discovery research [3]. Our understanding is that the surface representation serves as a complementary modeling method to atom- or residue-based geometric graph representations. Please see more experimental supporting evidence in our following response.
>
> [1] Duhovny et al., International workshop on algorithms in bioinformatics (2002): 185-200.
>
> [2] Shulman-Peleg et al., J. Mol. Biol. 339.3 (2004): 607-633.
>
> [3] Isert et al., arXiv preprint arXiv:2210.11250.
>
>
> **Comment 2**: The claim "... since the surface is approximately an isosurface of the molecular electron density field" is not convincing enough and may need additional justifications.
>
> **Response 2**: Thanks for pointing this out. We removed the statement about the isosurface of an electron density field in the revised manuscript to avoid confusion.
>
> **Comment 3**: Current experimental results are not convincing enough to show the significance and effectiveness of the proposed methods.
>
> **Response 3**: In the revised manuscript, we demonstrated the effectiveness of HMR on three experiments focusing on molecular systems of different sizes. These experiments showcase the representation power of HMR in regression, classification, and rigid docking tasks, which highlight the model versatility for downstream applications. Specifically,
> 1. The purpose of showing HMR performance on the QM9 regression tasks is to demonstrate its compatibility with molecules of different sizes. Atom-based graph neural networks seem to be a better representation for this task, yet our surface-based approach achieves comparable predictive power.
> 2. The HMR-powered rigid protein docking model outperforms current state-of-the-art deep learning model on this challenging task, and achieves comparable performance to conventional docking methods at a much faster inference speed. Therefore, the proposed surface-based method provides a viable solution to the significant protein-protein interaction problem.
> 3. In order to further prove the effectiveness of the proposed method, we performed a ligand-binding protein pocket classification task in comparison to MaSIF-ligand [4]. Our model outperforms the geodesic CNN-based MaSIF model, demonstrating its effectiveness in modeling protein-ligand interactions. Please see Section 5.2 in the revised manuscript for more details.
>
> [4] Gainza et al., Nature Methods 17.2 (2020): 184-192.
>
> **Comment 4**: For the binding task, the EquiDock can be a relatively weak baseline. The authors may want to include stronger baselines such as EquiBind.
>
> **Response 4**: Thanks for your suggestion. Due to the following reasons, we did not compare HMR with EquiBind at this time:
> 1. While EquiDock focuses on rigid protein-protein docking, EquiBind solves the flexible small-molecule docking problem. These two tasks are quite different: (a) the size of ligand is much smaller for EquiBind, whose flexibility has to be considered during docking; (b) the protein-ligand binding site is typically well-defined (i.e., a pocket), yet the protein-protein interface is much more challenging to identify.  Therefore, we could not directly compare HMR with EquiBind under current settings.
> 2. We agree that EquiDock might be a relatively weak baseline, yet it is still the state-of-the-art of deep learning-based protein-protein docking models. The main difference between EquiDock and EquiBind is the flexibility of the ligand molecule, where the critical point prediction modules are essentially the same. Interestingly, EquiBind exhibits much better performance for ligand docking tasks, whereas EquiDock has poor docking power for protein-protein interactions.
> 3. Practically, we are not able to perform the protein-ligand docking task given the response time frame. In addition, incorporating molecular flexibility requires recomputing the Laplace-Beltrami eigenfunctions upon conformational changes under the HMR framework, which is computationally intensive. We are actively exploring viable solutions (e.g., employing neural PDE solvers [5]).
>
> [5] Raissi et al., J. Comput. Phys. 378 (2019): 686-707.

---

> > ### Author Response · Authors · 2022-11-14
> > **Response to Reviewer tWAd - Part 2**
> >
> > **Comment 5**: There is a lack of details or comparisons on the method's computation complexity in terms of runtime and memory cost.
> >
> > **Response5**: Thanks for pointing this out. We added detailed computational resource consumption analysis in Appendix F-H. The preprocessing of molecular surface mesh is time-consuming and takes 80% of the total inference time for the protein docking task, yet the average inference time of HMR is comparable to that of EquiDock (5.8 sec vs. 3.9 sec per protein complex), and is much faster than ATTRACT (882.7 sec) and HDOCK (884.9 sec). Please see Table 2 and Table H.1 for more details.
> >
> > **Comment6**: It would be better if the authors include results on how increasing the dimensions (surface resolution) can improve the performance and increase the computation cost.
> >
> > **Response6**: Thanks for your suggestion.
> > 1. We reported the rigid protein docking model performance at different surface resolutions in Table H.6, where we observed a decreasing model performance with lower resolutions.
> > 2. Models with lower resolutions are slightly faster to train, yet the difference is not significant (since the cross attention module is the most computationally intensive step).
> > 3. Reducing the model resolution will also reduce data preprocessing time and data size. We empirically determined the resolution (i.e., the maximum eigenvalue to keep) for each experiment, where in general smaller molecules require fewer eigenfunctions.
> >
> > **Comment 7**: The proposed method itself is interesting and novel for geometric graph learning. I'm sure it can also be applied to many other scenarios than molecules.
> >
> > **Response 7**: Thanks for your compliment. In fact, we adopted relevant techniques for 3D computer vision from the geometry processing research community. The major distinction is that we need to additionally consider chemistry for molecular systems. From our experimental results, chemistry plays a dominant role in molecular systems, which is reasonable since molecules are not simply shapes.
> >
> > **Comment 8**: The proposed rigid protein docking framework has additional technical contributions.
> >
> > **Response 8**: Thanks for pointing this out. The fact that functional maps could nicely predict the protein complex structure is exciting and worth more investigation (e.g., interpreting the physical meaning of the learned surface functional correspondence). However, we later realized that the entire proposed HMR framework requires proper introduction due to the lack of prior research on this topic.

---

> > ### Comment · Reviewer_tWAd · 2022-11-14
> > **Some more questions**
> >
> > Thank you for your response! I have updated my evaluation as most of my concerns have been addressed.
> >
> > I have a few more questions persist regarding QM9 given your response:
> >
> > 1. To my understanding, the QM9 prediction relies on the precise geometry of molecules. Will some important details be missing when converting it to surface? It makes sense to me that the resolution is not critical to the binding task, but it maybe important for QM9. It'd be better if the authors include some experiment / empirical analysis on this (not necessarily during rebuttal).
> >
> > 2. Thank you for providing the comparisons on computational cost on the binding task. Can the authors also compare it for the QM9 task regarding different sizes? Specifically, if the surface representation consumes significantly less time/memory than existing 3D GNN approaches, we can say it's an efficient approximation of those approaches. That would add more contribution to the work.

---

> > > ### Author Response · Authors · 2022-11-17
> > > **Response to follow-up questions**
> > >
> > > Thanks for your follow-up questions, we are delighted to have more discussions with you. Please see our response below.
> > >
> > > **Comment**: To my understanding, the QM9 prediction relies on the precise geometry of molecules. Will some important details be missing when converting it to surface?
> > >
> > > **Response**: Yes - GNNs like SchNet and EGNN encode inter-atomic distance information and perform neural message passing to represent a molecule. Our surface-based model does not explicitly encode bonding (e.g., bond type, bond length, etc.) information, which is quite significant from the electronic structure theory perspective. Our model also does not have any atom-atom communications. These factors might explain why HMR underperforms state-of-the-art GNN models in QM9 tasks.
> > >
> > > Our understanding is that introducing the molecular Shape-DNA is essentially introducing a strong inductive bias to the deep learning model, where features are constrained on the surface manifold and can be analytically propagated according to the Laplace-Beltrami eigenfunctions. This might explain the success of the proposed model in the protein docking task, where training data is limited. It is possible that when training data is abundant, such strong inductive bias may limit the model expressiveness. We are actively exploring this interesting topic.
> > >
> > > We performed some resolution-tuning experiments on "HOMO" prediction (one of the QM9 regression tasks), the results are summarized in the table below. In addition, we would like to clarify that we only reduce the number of Laplace-Beltrami eigenfunctions for harmonic message passing, the molecular surface shape (i.e., the manifold) is not modified. Therefore, *lower resolution here means the model is only able to learn smooth functions over the surface for molecular representations*. It is possible that the underling molecular property could be expressed as the weighted sum of some smooth functions (learned by neural networks). In other words, reducing resolution does not necessarily lead to worse performance in this case.
> > >
> > > **Comment**: Can the authors also compare it for the QM9 task regarding different sizes? Specifically, if the surface representation consumes significantly less time/memory than existing 3D GNN approaches, we can say it's an efficient approximation of those approaches. That would add more contribution to the work.
> > >
> > > **Response**: Thanks for being considerate. We show the computational cost of the QM9 HOMO task in the table below. Here we also provide some analysis on model efficiency.
> > >
> > > For a molecule with $N$ atoms (in QM9 the average number of atoms is 18), a GNN model contains $N$ nodes and an $N \times N$ sparse adjacency matrix. Its molecular surface consists of $M$ vertices (on average 440 vertices with 47 eigenfunctions in our experiments). To realize harmonic message passing, we additionally need to include the $M \times M$ sparse mass matrix (for inverse Fourier transform, see Appendix A-B for details). From the brief analysis here, we see that the molecular graph representation is more lightweight for small molecules. However, computational efficiency might not be the bottleneck problem yet, as the training and inference time are reasonable in our experiments (see table below). Just like the evolution of Euclidean NNs (e.g., TFN -> SE(3)-Transformer -> EGNN -> SEGNN), more powerful and efficient models will hopefully be developed based on prior ones.
> > >
> > > **Table. Ablation studies on QM9 HOMO regression task.**
> > >
> > > |Max eigenvalue | Avg. num eigenvalues | HMR layers | Storage size (GB) | Num parameters | Training time/epoch (sec) | Inference time/epoch (sec) | MAE (meV) |
> > > |-|-|-|-|-|-|-|-|
> > > |1	|9	|0	|6.4	|54,785	|112	|14	|105.5|
> > > |1	|9	|1	|6.4	|105,089	|136	|14	|70.4|
> > > |1	|9	|3	|6.4	|205,697	|208	|16	|47.7|
> > > |0.2	|2	|6	|2.8	|356,609	|294	|16	|41.1|
> > > |0.4	|4	|6	|3.6	|356,609	|294	|16	|40.1|
> > > |0.6	|6	|6	|4.7	|356,609	|300	|16	|39.2|
> > > |0.8	|8	|6	|5.6	|356,609	|320	|18	|37.9|
> > > |1	|9	|6	|6.4	|356,609	|320	|18	|38|
> > > |2	|19	|6	|11	|356,609	|324	|18	|37|
> > > |3	|28	|6	|15	|356,609	|326	|18	|37|
> > > |4	|38	|6	|19	|356,609	|316	|18	|36.5|
> > > |5	|47	|6	|22	|356,609	|310	|18	|36.3|
> > >
> > >
> > > In our ablation studies, we first show the necessity to use harmonic message passing for accurate property regression. We fix the model resolution (maximum eigenvalue set to 1.0) and gradually increase the number of HMR layers. We observe a clear performance gain with more HMR layers. Then, we fix other model components and modify the resolution (by setting different maximum eigenvalues from 0.2 to 5.0). We see some performance gain with higher resolutions. Computational cost is also tabulated here for your reference.
> > >
> > > As discussed, more interesting questions await to be explored. We hope that this work could trigger curiosity of many readers, so that more scientists could work collaboratively to solve problems in this field.

---

### Author Response · Authors · 2022-11-14
**General Response**

We thank the reviewers for their comments and constructive suggestions. In summary, all reviewers acknowledged the novelty of the proposed HMR framework, and discussed the potential of applying this method to drug discovery research. Two common concerns include:
1. Providing more experimental support for the effectiveness of HMR on various tasks.
2. Analyzing the computational complexity of HMR in comparison to other models.

To address these concerns, we made the following major updates in the revised manuscript:
1. We conducted a protein pocket classification task in comparison to MaSIF-ligand [1], where HMR outperforms the geodesic CNN-based MaSIF model in predicting the metabolite-binding preference of a protein pocket using its surface features. Please see below and Section 5.2 in the revised manuscript for more details.

Table 1: Balanced accuracy of protein pocket classification.
| Model | Geom+Chem | Geom Only | Chem Only |
| - | -| - | - |
| MaSIF-ligand | 0.74 | 0.55 | 0.65 |
| HMR | 0.81 | 0.66 | 0.71 |

2. We provided a complete breakdown of computational resource consumption in three reported tasks, which includes preprocessing time, storage size, and inference time. Overall, HMR has a comparable protein docking inference time to EquiDock (5.8 sec vs. 3.9 sec per protein complex, including preprocessing time), which is two orders of magnitude faster than ATTRACT and HDOCK. Please check out Appendix F-H for more details.

Table 2: Rigid protein docking average inference time per protein complex.
| Model | |Time (sec) |
| - | - | - |
| HDOCK |  |884.9 |
| ATTRACT | | 882.7 |
| EquiDock | | 3.9 |
| HMR | | 5.8 |

In addition, we provided a comprehensive Appendix with more detailed theoretical derivations, computational simulation techniques, experimental setups, and result analyses. Our main objective is to spread out the idea and call for more research attention to the proposed method. To that end, we presented the paper in such a way that general readers could understand the basic ideas from the main text, while readers with interest can find the complete recipe to reproduce our work in the Appendix.

To facilitate reviewing the revised manuscript, we uploaded a PDF file where revisions are highlighted in blue.

[1] Gainza et al., Nature Methods 17.2 (2020): 184-192.

---

### Decision · Program_Chairs · 2023-01-20

**Decision:**

Accept: poster

**Justification For Why Not Higher Score:**

paper is based on techniques well-known in other field

**Justification For Why Not Lower Score:**

sufficiently interesting application and novelty for ML community

**Metareview: Summary, Strengths And Weaknesses:**

The paper uses spectral geometry techniques (that are common in the CG community) for molecular tasks, such as small molecule property prediction and protein docking. The reviewers appreciated the paper, and the authors provided an extensive rebuttal.
We recommend accepting.


**Note From Pc:**

if the above contains the word "oral" or "spotlight" please see: "oral" presentation means -> notable-top-5% and "spotlight" means -> notable-top-25%. As stated in our emails, we are disassociating presentation type from AC recommendations